## REPORT

# The Pex3–Inp1 complex tethers yeast peroxisomes to the plasma membrane

Georgia E. Hulmes[1], John D. Hutchinson[1], Noa Dahan[2], James M. Nuttall[1], Ellen G. Allwood[3], Kathryn R. Ayscough[3], and Ewald H. Hettema[1]

A subset of peroxisomes is retained at the mother cell cortex by the Pex3–Inp1 complex. We identify Inp1 as the first known plasma membrane–peroxisome (PM-PER) tether by demonstrating that Inp1 meets the predefined criteria that a contact site tether protein must adhere to. We show that Inp1 is present in the correct subcellular location to interact with both the plasma membrane and peroxisomal membrane and has the structural and functional capacity to be a PM-PER tether. Additionally, expression of artificial PM-PER tethers is sufficient to restore retention in inp1Δ cells. We show that Inp1 mediates peroxisome retention via an N-terminal domain that binds PI(4,5)P$_2$ and a C-terminal Pex3-binding domain, forming a bridge between the peroxisomal membrane and the plasma membrane. We provide the first molecular characterization of the PM-PER tether and show it anchors peroxisomes at the mother cell cortex, suggesting a new model for peroxisome retention.

## Introduction

Eukaryotic cells have evolved molecular mechanisms that control organelle size, number, and position. Molecular tethers are required for organelle positioning, multiplication, and establishment of interorganellar contact sites. The balance between organelle tethering and motility determines the intracellular distribution of organelles and their segregation during cell division (Fagarasanu et al., 2010). In *Saccharomyces cerevisiae*, correct peroxisome inheritance is achieved by the opposing processes of cortical anchoring in the mother cell and myosin-dependent transport toward the bud (Fagarasanu et al., 2006; Hoepfner et al., 2001). The Inp1–Pex3 tethering complex is required for peroxisome retention during cell division and for peroxisome positioning along the mother cortex (Fagarasanu et al., 2005; Knoblach and Rachubinski, 2013; Munck et al., 2009). As has been postulated for other organelles, yeast peroxisomes interact with many cellular structures, including the plasma membrane, ER, vacuole, mitochondria, and lipid bodies (Eisenberg-Bord et al., 2016; Knoblach and Rachubinski, 2019; Wu et al., 2019). Components of some interorganellar peroxisomal contact sites have recently been identified whereas others are still completely uncharacterized, including the plasma membrane–peroxisome (PM-PER) contact site (Shai et al., 2016).

In yeast cells, the plasma membrane and cortical ER (cER) are in very close proximity with up to 45% of the plasma membrane in contact with the cER network (Pichler et al., 2001; West et al., 2011). These areas of close contact between the two membranes create distinct, subcellular microenvironments and have been shown to have membrane contact sites with a third organelle,

the mitochondria (Lackner et al., 2013). Cortical mitochondrial anchorage occurs via the mitochondrial ER cortex anchor (MECA), a multisubunit protein complex (Lackner et al., 2013). The core component of MECA is Num1, a cortex-associated protein that binds to the mitochondrial outer membrane and the plasma membrane via two distinct lipid-binding domains (Lackner et al., 2013). Although MECA is reported to be a tether among three organelle membranes, the molecular mechanisms for MECA association with the ER remain poorly understood.

The plasma membrane has been further implicated in mitochondrial tethering via site-specific mitochondrial anchorage at the mother cell tip via Mfb1 (Manford et al., 2012; Pernice et al., 2016). As the plasma membrane has been shown to be required for cortical retention and correct spatial distribution of mitochondria, this raises the possibility that organelle retention via the plasma membrane could also occur in the cases of other cortically associated organelles.

The existence of PM-PER contact sites has been reported (Shai et al., 2018), but as yet, these remain uncharacterized. Previously, peroxisomes have been reported to be retained via Inp1 and the cER; however, a recent study suggests that additional components are required for peroxisome tethering to the cell cortex (Knoblach and Rachubinski, 2019; Knoblach et al., 2013). Through detailed analysis of the molecular function of Inp1, we have uncovered a novel role for Inp1 as a PM-PER tether and conclude that tethering of peroxisomes to the plasma membrane is required for peroxisome retention.

[1]Department of Molecular Biology and Biotechnology, University of Sheffield, Sheffield, England, UK; [2]Department of Molecular Genetics, Weizmann Institute of Science, Rehovot, Israel; [3]Department of Biomedical Science, University of Sheffield, Sheffield, England, UK.

Correspondence to Ewald H. Hettema: e.hettema@sheffield.ac.uk.

We describe a conserved Pex3-binding motif in the C-terminal region of Inp1. This motif bears a striking resemblance to the Pex3-binding site present on Pex19, and we present both in vitro and in vivo evidence that Pex19 and Inp1 can compete for binding to Pex3. In addition, we show that the N-terminal 100 amino acids of Inp1 localize to the plasma membrane, bind to phosphatidylinositol 4,5-bisphosphate ($PI(4,5)P_2$) and, when artificially attached to the peroxisomal membrane, restore retention by relocating peroxisomes to the cell periphery in *inp1Δ* cells. Furthermore, we show that this region of Inp1 is also necessary for retention. We demonstrate that Inp1 is a component of the PM-PER tether according to the guidelines that define the prerequisites for categorizing a protein as a tether (Eisenberg-Bord et al., 2016), as (1) it is present in the correct subcellular location, apposed to the peroxisome and plasma membranes; (2) Inp1 has the structural capacity to form a PM-PER tether by binding the peroxisomal membrane via a C-terminal Pex3-binding domain and binding the plasma membrane via an N-terminal domain that binds $PI(4,5)P_2$; and (3) Inp1 has the functional capacity to form a PM-PER tether, as an artificial tether that links the peroxisomal membrane to the plasma membrane restores peroxisome retention in *inp1Δ* cells and overexpression of this minimal tether increases the number of PM-PER contact sites. These observations qualify the Inp1 as the first PM-PER tether and support a new model for peroxisome retention.

## Results

### Inp1 is in the correct subcellular location to be a PM-PER tether

During cell division, a subset of peroxisomes are retained by cortical anchoring in the mother cell (Fagarasanu et al., 2005; Hoepfner et al., 2001; Knoblach and Rachubinski, 2019; Knoblach et al., 2013). Though the role of the cER in peroxisome retention has been reported, the role of the plasma membrane remains unexplored. We first visualized peroxisome distribution by expressing the peroxisomal matrix marker HcRed-PTS1 (HcRed with a type 1 peroxisome targeting signal) in two mutants where the cER is disrupted. *rtn1/rtn2/yop1Δ* cells do not form cER tubules but instead produce large ER sheets, leaving parts of the plasma membrane free of ER (Voeltz et al., 2006). We also looked at the mutant *scs2/scs22/ist2/tcb1/tcb2/tcb3Δ* (ER-PM tetherΔ), where the cER is almost completely collapsed from the cell cortex (Manford et al., 2012). In both strains, disruption of the cER does not appear to affect peroxisome distribution, with peroxisomes retained in mother cells as in WT. Deletion of *INP1* in these strains results in a typical *inp1Δ* phenotype with most peroxisomes present in the bud (Fig. 1, A and B).

To further visualize peroxisome localization when the cER is disrupted, we coexpressed the ER marker Sec63-mRFP, the plasma membrane marker GFP-Sso1, and BFP-PTS1 in *rtn1/rtn2/yop1Δ* cells. Peroxisomes in this mutant were normally distributed, present at the mother cell periphery, and often seen at the exposed ends of ER sheets. 46% of budding cells contained at least one peroxisome at the periphery in areas devoid of ER (*n* = 170; Fig. 1, C and E). To further visualize the spatial relationships

among peroxisomes, ER, and the plasma membrane, line-scan analyses were done, plotting relative fluorescent intensity versus distance. These confirmed an overlap in the signal of peroxisomal foci and the plasma membrane indicative of PM-PER contact sites.

In ER-PM tetherΔ cells, despite a lack of cER, approximately half of cells (50.3%, *n* = 150) contained at least one peroxisome, which was peripherally located, even in the absence of ER (Fig. 1, D and E). When *INP1* was also deleted from the ER-PM tetherΔ strain, peroxisomes were no longer localized at the periphery but were still observed in close proximity to internal cellular ER structures in buds (Fig. 1 D).

Tether proteins must reside in a defined location at the contact site (Eisenberg-Bord et al., 2016). We explored the subcellular localization of Inp1 by coexpressing Inp1-mCherry, Sec63-GFP, and BFP-PTS1 in *rtn1/rtn2/yop1/inp1Δ* cells (Fig. 1, F and G). Inp1-mCherry rescues the peroxisome retention defect, and where peripheral peroxisomes could be seen localized to the exposed ends of ER sheets or in areas free of ER, Inp1 foci always partially colocalized with peroxisomal matrix marker foci on the side of peroxisomes proximal to the plasma membrane. Where peroxisomes were seen "sandwiched" between the ER and the cell periphery (Fig. 1 G, image 3), peroxisomal matrix foci were consistently located between Inp1 foci and the ER. Inp1 was again proximal to the cell periphery and distal from ER sheets. Line-scan analyses confirmed that the profiles of Inp1-mCherry were slightly juxtaposed to those of the peroxisomal matrix foci but the peak intensities of Inp1-mCherry and the ER were spatially resolved (Fig. 1 G). Where Inp1-mCherry was coexpressed with BFP-PTS1 and the plasma membrane marker GFP-Sso1, Inp1-mCherry again partially overlapped with peroxisomal matrix foci proximal to the cell periphery and colocalized with GFP-Sso1 at the plasma membrane (Fig. 1 H). From these observations, we conclude that Inp1 is closely apposed to the peroxisomal and plasma membranes and hence is present in the correct subcellular location to potentially be a PM-PER tether.

### The C-terminal domain of Inp1 is required for localization to peroxisomes and binding to Pex3

A tether must have structural capacity to bind to the opposing membranes of two organelles (Eisenberg-Bord et al., 2016). We first set out to characterize exactly how Inp1 binds to the peroxisomal membrane. Pex3 is required for Inp1 association with peroxisomes and directly binds to Inp1 (Knoblach et al., 2013; Munck et al., 2009). To elucidate which regions of Inp1 are required for association with peroxisomes, we expressed a library of Inp1-GFP truncations in *inp1Δ* or *pex3Δ* cells. As expected, when expressed in *inp1Δ* cells, full-length Inp1-GFP was present in punctate structures that colocalized with the peroxisomal marker and was cytosolic in *pex3Δ* cells where peroxisomes are absent (Figs. 2 A and S1 A). This localization pattern was observed for N-terminal truncations of Inp1-GFP up to and including amino acids 300–420. However, when Inp1 was further truncated (amino acids 315–420), it was no longer associated with peroxisomes (Figs. 2 A and S1 A). Further fine mapping revealed that the minimal region that still associated with peroxisomes comprises residues 311–370 (Fig. S1 B). A genome-wide

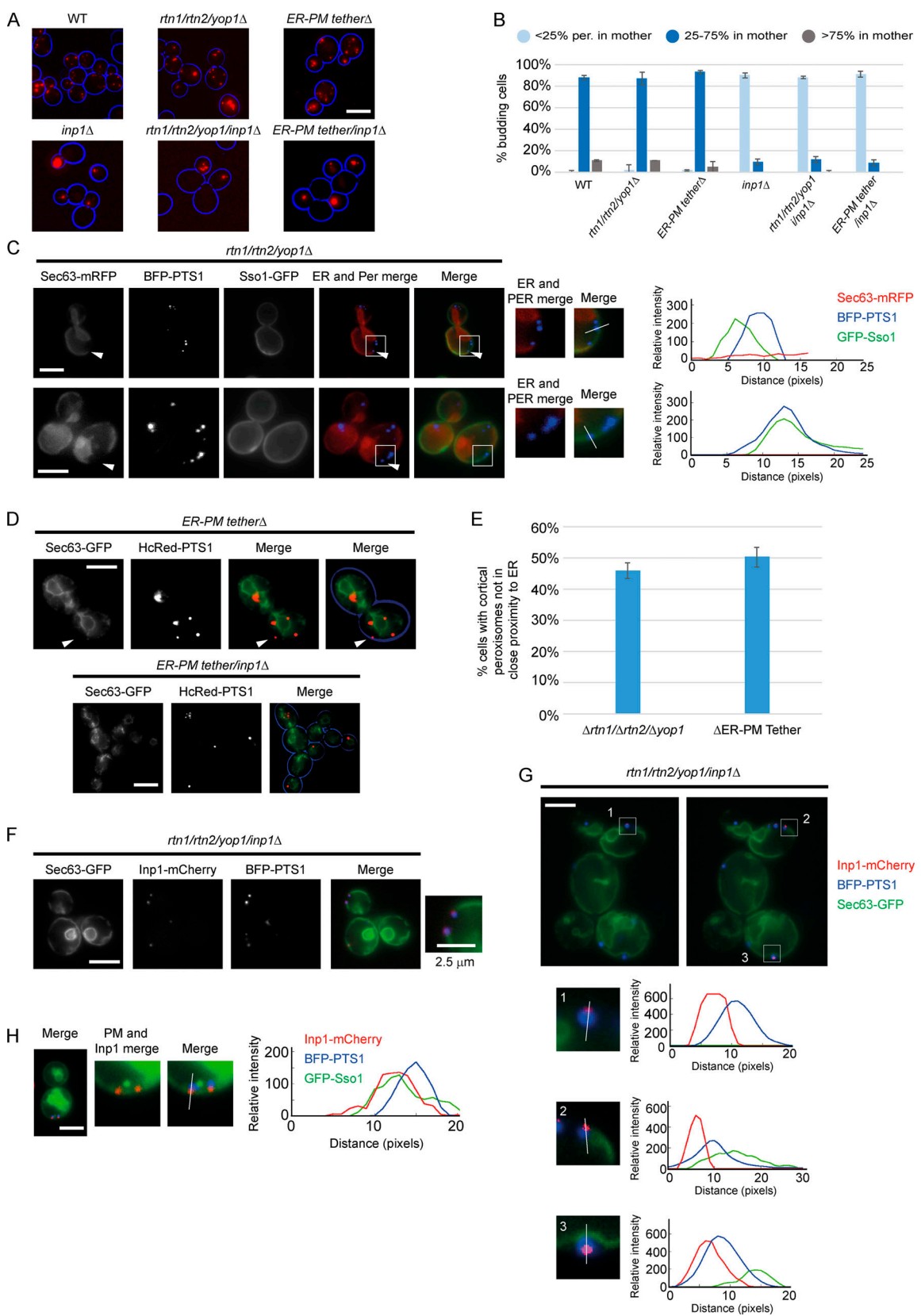

Figure 1. **Inp1 is present at the correct subcellular location to be a PM-PER tether. (A)** The peroxisomal matrix marker HcRed-PTS1 was constitutively expressed in WT, *rtn1/rtn2/yop1Δ*, ER-PM tetherΔ, *inp1Δ*, *rtn1/rtn2/yop1/inp1Δ*, and ER-PM tether/*inp1Δ* cells. Cells were examined by epifluorescence microscopy. **(B)** The peroxisome distribution of the strains described in A were analyzed. per., peroxisomes. Over 100 budding cells were analyzed for each strain. Three independent experiments were performed. Error bars represent SEM. **(C)** GFP-Sso1, Sec63-mRFP, and BFP-PTS1 were coexpressed in *rtn1/rtn2/yop1Δ*

cells and imaged using epifluorescence microscopy. Single focal planes are shown from the center of the z-stack. The arrowhead on the upper panel highlights a peroxisome (Per) present at the end of an ER sheet. The arrowhead on the lower panel highlights peroxisomes at the cell periphery, where there is no visible ER. The boxed areas are magnified. The graphs show relative fluorescence intensities of the ER (red), the peroxisomal matrix (blue), and the plasma membrane (green) along a line drawn through the center of the peroxisomal foci. **(D)** Sec63-GFP and HcRed-PTS1 were coexpressed in ER-PM tetherΔ or ER-PM tether/ *inp1Δ* cells and examined by epifluorescence microscopy. The arrowhead highlights an example of a peripheral peroxisome where there is no visible ER. A single focal plane is shown from the center of the z-stack. **(E)** Budding *rtn1/rtn2/yop1Δ* and ER-PM tetherΔ cells were scored for the presence of at least one peroxisome present at the cell periphery, but not associated with visible ER. Over 100 budding cells were quantified for each strain. Three independent experiments were performed. Error bars represent SEM. **(F and G)** Sec63-GFP, Inp1-mCherry and BFP-PTS1 were coexpressed in *rtn1/rtn2/yop1/inp1Δ* cells and imaged using epifluorescence microscopy. In F, a single focal plane is shown. In G, two single, consecutive z-stack layers of the image are shown. The boxed areas are magnified. The graphs show relative fluorescence intensities of the ER (green), the peroxisomal matrix (blue), and Inp1 (red) along a line drawn through the center of the peroxisomal foci. **(H)** GFP-Sso1, Inp1-mCherry, and BFP-PTS1 were coexpressed in *rtn1/rtn2/yop1/inp1Δ* cells and imaged using epifluorescence microscopy. A single focal plane is shown. The graph shows relative fluorescence intensities of the plasma membrane (green), the peroxisomal matrix (blue), and Inp1 (red) along a line drawn through the center of the peroxisomal foci. Scale bars, 5 µm.

yeast two-hybrid screen using the Pex3 cytosolic domain (40–441; Motley et al., 2012) identified 14 independent Inp1 DNA clones that encode 10 different Inp1 protein fragments. The smallest fragment that interacts comprises amino acids 300–378, and this region was common to all hits (Fig. S1 B). This region encompasses the minimal truncation of Inp1 that still localizes to peroxisomes and identifies this C-terminal region as being important for Pex3 binding.

We performed an in vitro binding assay with *Escherichia coli*–expressed maltose-binding protein (MBP) fusions of the N- or C-terminus of Inp1 (amino acids 1–280 and 281–420, respectively) and His-Pex3 (40–441; Fig. 2 B). A direct interaction was found between His-Pex3 and the C-terminus of Inp1 (281–420), reinforcing that the C-terminal region of Inp1 is required for binding directly to Pex3. It has been reported that both the N- and C-terminal part of Inp1 bind directly to Pex3 in vitro (Knoblach et al., 2013). We were unable to reproduce specific binding of the N-terminal part of Inp1 to Pex3 under our assay conditions. A closer look at the amino acid sequences of Inp1 orthologues highlighted a highly conserved leucine-based motif flanked by negative charged residues within the region of Inp1 that is necessary for localization to peroxisomes and binding to Pex3, corresponding to amino acids 312–316 (Fig. 2 C).

### The Pex3 binding site of Inp1 resembles the Pex3 binding site of Pex19

The Pex3–Pex19 interaction has been well characterized and a "leucine triad" motif in human Pex19 has been identified as being critical for Pex3 binding and peroxisome biogenesis (Agrawal et al., 2017; Sato et al., 2010; Schmidt et al., 2010, 2012). This designated Pex3 binding motif of Pex19 (which we refer to as LXXLL) is highly conserved throughout eukaryotes including *S. cerevisiae* and bears a striking similarity to the motif we have identified in Inp1 (Fig. 2 D).

Substitution of the leucine residues in the LXXLL motif in human Pex19 with alanines results in decreased affinity for Pex3 (Sato et al., 2010). Mutation of the leucine triad in *S. cerevisiae* Pex19 (ScPex19) results in mislocalization of HcRed-PTS1 to the cytosol (Fig. S1 C), although the GFP-ScPex19 mutants are well expressed (Fig. S1 D). This confirms that the LXXLL motif is also required for Pex19 function in *S. cerevisiae*.

The Inp1 LXXLL motif was subsequently targeted for mutagenesis. For this, we expressed Inp1-GFP under control of its endogenous promoter in *inp1Δ* cells. As previously described,

peroxisomes in mother cells are labeled more intensely with Inp1-GFP than those in the buds (Knoblach et al., 2013). Deletion of the full LXXLL motif results in a loss of peroxisomal localization and a failure to rescue retention (Fig. 2 E). Subsequently, an alanine scan of the motif region established that it is the leucine residues that are important for the peroxisomal localization and function of Inp1 (Fig. 2 E; and Fig. S1, E and F). Although some mutants were no longer localized to peroxisomes and hard to detect with epifluorescence microscopy, Western blot analysis shows that all of these mutants are expressed well (Figs. 2 F and S1 G). Since the Inp1 LXXLL deletion mutant and the Inp1 LL>AA mutant had the strongest effect on localization and function, we tested these mutants in an in vitro binding assay. Both mutants were strongly compromised in direct binding to His-Pex3 (Fig. 2 G). From this, we conclude that Inp1 contains an LXXLL motif that is required for its function, as it is involved in binding directly to Pex3 and the localization of Inp1 to peroxisomes. During the course of our study, an independent study reported a random Inp1 mutant in this motif (LXXLP). This mutant also blocked recruitment of Inp1 to peroxisomes and interfered with Inp1 function (Knoblach and Rachubinski, 2019).

### Pex19 and Inp1 compete for binding Pex3, and this is dependent on their LXXLL motifs

Since both Inp1 and Pex19 have an LXXLL motif required for their respective functions and both bind to Pex3 to perform these functions, it is a logical assumption that they compete for binding to Pex3. A total lysate of *E. coli* cells expressing His-Pex3 was preincubated with a total lysate of *E. coli* cells expressing MBP-Inp1 to allow Pex3 to bind Inp1. This unpurified mixture was then poured over GST-Pex19 fusions immobilized on glutathione Sepharose beads. The presence of MBP-Inp1 prevents His-Pex3 from binding to immobilized GST-Pex19, demonstrating that Inp1 competes with Pex19 for binding Pex3 in vitro. When an *E. coli* lysate containing MBP was used instead of MBP-Inp1 during the preincubation, no competition was observed (Fig. 3 A). The competition assay was then repeated with MBP fusions of the N- or C-termini of Inp1 (1–280 or 281–420, respectively), and only the C-terminus was found to prevent GST-Pex19 from binding to His-Pex3 (Fig. 3 B). This suggests that a Pex3-Inp1 C complex had formed during the preincubation (as also observed in Fig. 2 B). To confirm this, the unbound fraction of the competition assay was further analyzed and revealed the presence of Pex3-Inp1 C complexes (Fig. S2 A). These

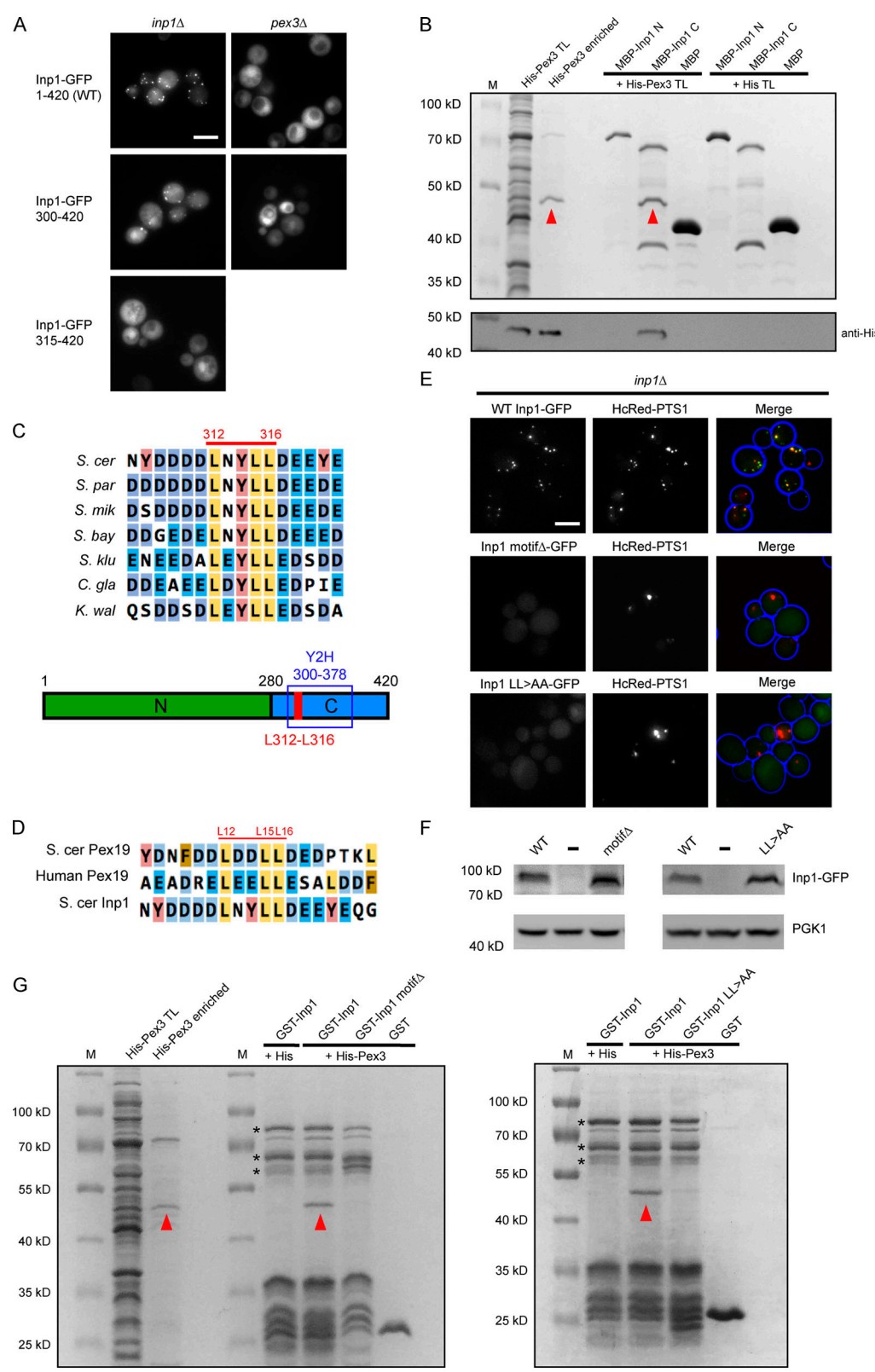

Figure 2.   **A C-terminal motif in Inp1 interacts with Pex3. (A)** Truncations of Inp1-GFP expressed under the control of the *GAL1/10* promoter in *inp1Δ* or *pex3Δ* cells were examined by epifluorescence microscopy. Scale bar, 5 µm. **(B)** *E. coli*–expressed MBP-Inp1 N (1–280) or MBP-Inp1 C (281–420) were bound to amylose beads and incubated with a total lysate of *E. coli* expressing either His-Pex3 (40–441) or His-tag only. After extensive wash steps, bound fractions were analyzed by SDS-PAGE followed by Coomassie staining and Western blotting. A lane was included with enriched His-Pex3 as a control. M, molecular weight marker; TL, total lysate. Arrowheads indicate His-Pex3. **(C)** Multiple sequence alignment of the conserved motif region in Inp1 orthologues of species closely related to *S. cerevisiae* (*S. cer*), *Saccharomyces paradoxus* (*S. par*), *Saccharomyces mikatae* (*S. mik*), *Saccharomyces bayanus* (*S. bay*), *Saccharomyces kluyveri* (*S. klu*), *Candida glabrata* (*C. gla*), and *Kluyveromyces waltii* (*K. wal*). The residues numbered in red highlight the conserved leucine motif in *S. cer* Inp1. The schematic

diagram shows an overview of the hypothetical Inp1 domain structure. Green, N-terminal domain 1–280; blue, C-terminal domain 281–420; red, conserved leucine motif. Y2H box, the smallest region of Inp1 found to bind to Pex3 in a yeast two-hybrid screen. **(D)** Multiple sequence alignment of *S. cer* Pex19, human Pex19, and *S. cer* Inp1. The leucine-based motif residues numbered in red correspond to *S. cerevisiae* Pex19. **(E)** Inp1-GFP, a leucine motif deletion (L312_L316del; motifΔ), and a LL 315,316 AA mutant (LL>AA) were expressed in *inp1Δ* cells under control of the *INP1* promoter with the peroxisomal matrix marker HcRed-PTS1. Scale bar, 5 μm. **(F)** Immunoblot analysis of Inp1-GFP mutants. (-), empty plasmid control. *PGK1*, loading control. **(G)** GST-Inp1, GST-Inp1 motifΔ, GST-Inp1 LL>AA and GST were bound to glutathione Sepharose and incubated with a lysate of *E. coli* expressing either His-Pex3 or His-tag only. After extensive wash steps, bound fractions were analyzed by SDS-PAGE followed by Coomassie staining. Arrowheads indicate His-Pex3. Asterisks indicate multiple GST-Inp1 fragments.

results reinforce that the C-terminus of Inp1 binds to Pex3, and a motif in this region directly competes with Pex19 for a binding site on peroxisomal Pex3.

Overexpression of Inp1 leads to mislocalization of a peroxisomal matrix marker to the cytosol (Fig. S3 of Knoblach et al., 2013). This is compatible with the in vitro competition we observed between Pex19 and Inp1 for binding to Pex3, as *pex19Δ* cells lack peroxisomes (Götte et al., 1998). We investigated whether the lack of matrix protein import is caused by an impairment in Pex19 activity or by other aspects of peroxisome dynamics. Mislocalization of a peroxisomal matrix protein can be the result of a block in matrix protein import or peroxisomal membrane biogenesis, excessive peroxisome degradation, or missegregation of peroxisomes.

When Inp1 is constitutively overexpressed under the control of the *TPI1* promoter in WT cells, GFP-PTS1 was mislocalized to the cytosol and the peroxisomal membrane marker Pex11-GFP no longer displayed a punctate pattern but rather displayed a tubular network (Fig. 3 C). These phenotypes resemble those found in *pex3Δ* and *pex19Δ* cells, where matrix proteins and Pex11 mislocalize to the cytosol and mitochondria, respectively (Motley et al., 2015), suggesting that typical peroxisomal membranes are absent upon Inp1 overexpression. This is not an effect of increased pexophagy (Fig. S2 B). Overexpression of Inp1 LL>AA did not affect localization of GFP-PTS1 and Pex11-GFP (Fig. 3 C).

The absence of peroxisomes in cells overexpressing Inp1 is initially a result of overretention in mother cells with buds failing to inherit peroxisomes (Fig. S1 B of Munck et al., 2009). Cells that fail to inherit peroxisomes are able to reform peroxisomes (Motley and Hettema, 2007), a process that relies on Pex3 and Pex19 (Hoepfner et al., 2005). However, in Inp1-overexpressing cells, this appears to be blocked, as no peroxisomes are present. To directly test this, we used an assay for de novo peroxisome formation based on the mating of *pex3Δ* and *pex19Δ* cells (Motley and Hettema, 2007). Constitutive overexpression of Inp1 blocked de novo formation upon mating (Fig. 3 D). Again, overexpression of Inp1 LL>AA did not block peroxisome formation as in any of our in vivo assays (Fig. 3, C and D). Our data support the idea that Inp1 competes with Pex19 for binding to Pex3 dependent on its LXXLL motif.

## The N-terminal 100 amino acids of Inp1 are necessary and, when associated with peroxisomes, sufficient for peroxisome retention

After characterizing the structural capacity of Inp1 to bind to peroxisomes via Pex3, we set out to create a minimal version of Inp1 in order to identify the region that is required for peroxisome retention independent of its binding to Pex3. First, we found that the N-terminal 100 amino-acid residues are required

for function as Inp1 100–420-GFP does not restore peroxisome retention in *inp1Δ* cells (Figs. 4 A and S3 A). We subsequently fused truncations of Inp1 to GFP-Pex15, a tail-anchored peroxisomal membrane protein with a cytosol-facing globular domain (Elgersma et al., 1997), and expressed them under control of the *INP1* promoter in *inp1Δ* cells. When associated with peroxisomes, the N-terminal 100 amino-acid residues of Inp1 were sufficient to retain peroxisomes at the mother cell cortex. We did note that peroxisomes in mother cells could frequently be observed clustering close to the bud neck (Fig. 4 B; and Fig. S3, B and C). The functional capacity of Inp1 to be a PM-PER tether was further tested by investigating if the *inp1Δ* phenotype could be rescued by a completely artificial tether. For this, we used Num1, a mitochondria-plasma membrane tether that interacts with the plasma membrane via its highly selective PI(4,5)P$_2$ binding pleckstrin homology (PH) domain (Yu et al., 2004; Tang et al., 2009). We fused this PH domain (amino acids 2,563–2,692) to Pex15 spaced by a GFP moiety. This artificial PM-PER tether also restored peroxisome retention in *inp1Δ* cells and resembled our Inp1 minimal tether (Fig. 4 B). As the loss of a tether should be rescued by expression of synthetic components that are able to tether the respective membrane (Eisenberg-Bord et al., 2016), we conclude that Inp1 has the functional capacity to be a PM-PER tether and that the N-terminal region of Inp1 is necessary and, when associated with peroxisomes, sufficient for retention of peroxisomes to the mother cell cortex.

## Inp1 is a component of the PM-PER contact site

If the overexpression of a protein results in an overall cellular increase of the contact site area between membranes, then this is a good indicator that this protein contributes to interorganelle tethering (Eisenberg-Bord et al., 2016). As overexpression of Inp1 results in a loss of peroxisomes (see Fig. 3, C and D), we had to take an alternative approach. We overexpressed the minimal peroxisomal tether (Inp1 1–100-GFP-Pex15). As expected, this results in overretention of peroxisomes in mother cells but does not affect peroxisome biogenesis (Fig. S3 D). We then overexpressed the minimal tether in a series of split Venus reporter strains that label peroxisomal membrane contact sites with a variety of organelles (Shai et al., 2018) and a PM-PER reporter comprising the peroxisomal membrane protein Pex11 and the plasma membrane protein PMP3. The number of fluorescent puncta representing peroxisomal contact sites with ER, mitochondria, and vacuoles was unaffected by this overexpression (Fig. 4 C). On the other hand, the number of PM-PER reporter puncta in the cell population increased with overexpression of the minimal tether. Both the frequency of cells with PM-PER reporter signal and the number of puncta per cell increased (Fig. 4, C and D).

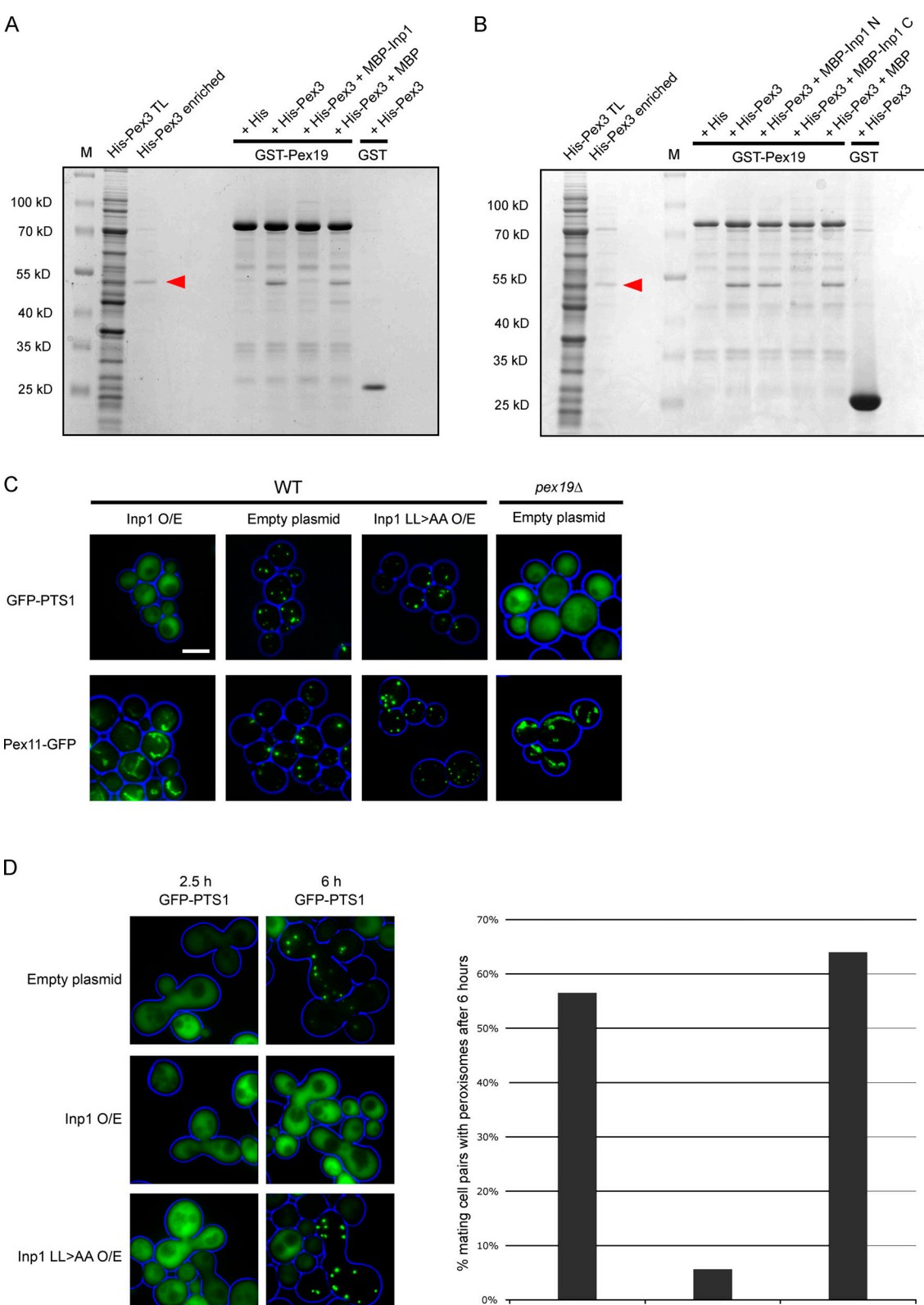

Figure 3. **Inp1 is able to compete with Pex19 for Pex3 binding. (A)** *E. coli* total lysates expressing His-Pex3 (40–441) or His-tag only were mixed with buffer only or with *E. coli* total lysates containing either MBP-Inp1 or MBP. These mixtures were then incubated with GST-Pex19 or GST immobilized on glutathione Sepharose beads. After extensive washing, including a 0.5 M KCl wash, bound fractions were analyzed by SDS-PAGE and Coomassie staining of the gel. A lane was included with enriched His-Pex3 as control. M, molecular weight marker. Arrowhead indicates His-Pex3. **(B)** The in vitro binding assay was done the same as described in A, but instead of MBP-Inp1, N- and C-terminal truncations of MBP-Inp1 (1–280 and 281–420, respectively) were used. **(C)** Epifluorescence microscopy images of WT cells overexpressing Inp1 or Inp1 LL>AA under control of the *TPI1* promoter with either GFP-PTS1 or the peroxisomal membrane

marker Pex11-GFP. An empty plasmid was used as a control. *pex19Δ* cells were transformed with GFP-PTS1 or Pex11-GFP to show the location of these proteins in the absence of typical peroxisomal structures. Scale bar, 5 μm. **(D)** *pex3Δ* (*MATa*) cells expressing GFP-PTS1 were mated with *pex19Δ* (*MATα*) cells over-expressing (O/E) either Inp1 or Inp1 LL>AA (under the *TPI1* promoter) or an empty plasmid. The cells were imaged after 2.5 or 6 h to check for de novo formation of peroxisomes. The graph presents quantitative analysis of the mating assay. Mating cell pairs forming zygotes were scored for the presence or absence of peroxisomes after 6 h. At least 100 mating cell pairs were quantified for each assay.

To visualize the subcellular localization of the minimal tether, it was coexpressed in *rtn1/rtn2/yop1Δ* cells with BFP-PTS1 and Sec63-GFP. The minimal tether was always present on the peripheral side of the ER, proximal to the plasma membrane, and in ~49% of cells appeared in areas devoid of cER (n = 150 cells; Fig. 4, E and F; and Fig. S3 E). This is comparable with the 46% of *rtn1/rtn2/yop1Δ* cells that had peripheral peroxisomes in the absence of visible cER when expressing WT Inp1 (Figs. 1 and S3 E). Again, peroxisomes appeared "sandwiched" between cER and the cell periphery, with the Inp1 minimal tether foci consistently juxtaposed to peroxisomal matrix foci, closer to the cell periphery, and spatially resolved from the ER (Fig. 4 F). The N-terminal domain of Inp1, when retaining peroxisomes as a minimal tether, is in the correct subcellular location to be associated with the plasma membrane.

### The N-terminus of Inp1 interacts directly with the plasma membrane

The subcellular localization of the N-terminal region was determined by fusing the first 100 amino acids of Inp1 to GFP under control of the inducible *GAL1/10* promoter. The protein was found in the cytosol and the nucleus but also showed obvious plasma membrane localization (Fig. 5 A). Nuclear localization has previously been observed with GFP fusions of plasma membrane–associated lipid-binding domains (Yu et al., 2004), although the reason for this has not been established.

To test the capability of Inp1 to bind to the plasma membrane in vitro, lipid-binding assays were performed. Purified *E. coli*–expressed MBP-Inp1 was incubated with liposomes prepared from bovine brain lipid extract (Folch fraction I). Binding was assessed by cosedimentation with ultracentrifuged liposomes. This established that MBP-Inp1 is able to bind to lipids in vitro (Fig. 5, B and C). To further characterize the Inp1–lipid interaction, synthetic liposomes were made up of phosphatidylcholine (PC), phosphatidylethanolamine (PE), and 22% phosphatidylinositol 4-phosphate (PI(4)P), PI(4,5)P$_2$, or phosphatidylserine (PS), as described in Materials and methods (Fig. 5 D–G). MBP-Inp1 showed clear binding to PC/PE liposomes containing PI(4,5)P$_2$ but failed to interact with PC/PE liposomes alone or when supplemented with PS or PI(4)P. This is a significant finding, as PI(4,5)P$_2$ is predominantly localized to and synthesized at the yeast plasma membrane (Vernay et al., 2012; Yu et al., 2004). To ensure that MBP-Inp1 binding to PI(4,5)P$_2$ liposomes is not simply due to nonspecific, charge-dependent interactions, the lipid-binding assay was repeated under high-salt conditions (Figs. S3 F and 5 G). Loss of interaction at a higher salt concentration indicates that the interaction between a protein and lipid is mainly mediated through nonspecific electrostatic interactions (Lemmon, 2008; Zhao and Lappalainen, 2012). MBP-Inp1 showed clear binding to PC/PE

liposomes containing PI(4,5)P$_2$ even in the presence of high salt (370 mM), indicating that besides electrostatic interactions, MBP-Inp1 has additional mechanisms of binding PI(4,5)P$_2$. Samples with purified MBP-Inp1 or GST-Inp1 contain truncated fragments (Knoblach and Rachubinski, 2013; Munck et al., 2009), which was informative, as some truncations were able to bind to liposomes (Fig. 5 D). As our MBP fusion proteins were N-terminally tagged, and since the molecular weight of MBP is 42.5 kD, we deduced that the N-terminal region of Inp1 is able to bind lipids. N- and C-terminal truncations of Inp1 were expressed in *E. coli* and assayed for their ability to interact with PI(4,5)P$_2$. The N-terminal truncations (1–280 and 1–100) bind PI(4,5)P$_2$ liposomes, in contrast to Inp1 281–420 (Fig. 5, H and I). We conclude that Inp1 is a lipid-binding protein able to bind PI(4,5)P$_2$, but not the related PI(4)P. The N-terminal 100 amino-acid residues are sufficient for PI(4,5)P$_2$ binding in vitro.

To determine whether cellular PI(4,5)P$_2$ levels would affect the plasma membrane localization of Inp1 1–100-GFP, a *sac1Δ* mutant was used. In *sac1Δ* cells, PI(4)P accumulates to levels 20-fold higher than normal, and PI(4,5)P$_2$ levels are depressed by ~75–80% (Foti et al., 2001; Hughes et al., 2000). Previously, it has been shown that other PI(4,5)P$_2$-binding GFP fusion proteins are less plasma membrane localized in *sac1Δ* cells (Yu et al., 2004). Likewise, we observed a visible reduction in the plasma membrane localization of Inp1 1–100-GFP in *sac1Δ* cells (Fig. 5 J). Line-scan analyses of relative fluorescence intensity versus distance confirmed the loss of Inp1 1–100 GFP fluorescent signal enrichment at the plasma membrane in *sac1Δ* cells compared with WT cells (Fig. S3 G). We conclude that Inp1 is a modular protein with an N-terminal lipid-binding domain and a C-terminal Pex3-binding region. This illustrates that Inp1 has the structural capacity to form a PM-PER tether by directly binding to PI(4,5)P$_2$ on the plasma membrane and Pex3 on the peroxisomal membrane.

## Discussion

Together, our experiments identify Inp1 as a component of the PM-PER tether. The N-terminal 100 amino acids of Inp1 localize to the plasma membrane, bind to PI(4,5)P$_2$ liposomes, and can act as a minimal tether which is necessary and, when attached to the peroxisomal membrane, sufficient for peroxisome retention at the cell periphery. An artificial tether composed of the plasma membrane–binding PH domain of Num1 artificially attached to peroxisomes is sufficient to restore peroxisome retention in *inp1Δ* cells. Additionally, when Inp1 and the Inp1 minimal tether are expressed in cER mutants, they localize to the cell periphery spatially resolved from the ER, on the side of peroxisomal foci proximal to the plasma membrane. Our findings support a

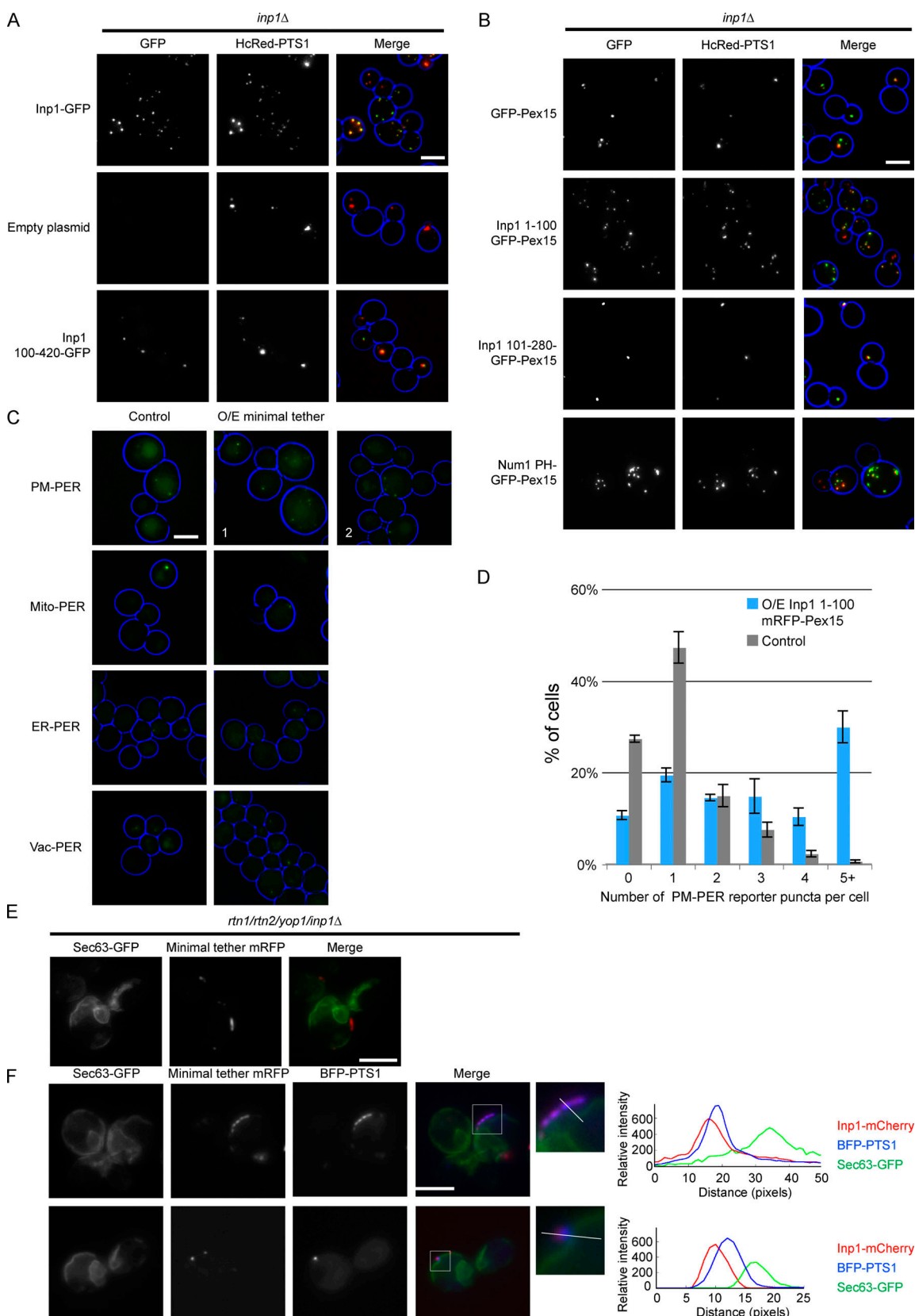

Figure 4. **Inp1 has the structural and functional capacity to be a PM-PER tether. (A)** The Inp1-GFP truncation 100–420 was expressed under control of the *INP1* promoter in *inp1Δ* cells with the constitutively expressed HcRed-PTS1. Cells were examined by epifluorescence microscopy. **(B)** Truncations of Inp1 or the PH domain of Num1 (amino acids 2,563–2,692) fused to GFP-Pex15 were expressed under control of the *INP1* promoter in *inp1Δ* cells with the constitutively expressed HcRed-PTS1 and examined by epifluorescence microscopy. **(C)** The Inp1 1–100 truncation was fused to mRFP-Pex15, and this minimal tether was

overexpressed under the control of the *TPI1* promoter in different peroxisome contact site split reporters. ER-PER, ER–peroxisome; Mito-PER, mitochondria–peroxisome; Vac-PER, vacuole–peroxisome. Cells were examined by epifluorescence microscopy for the presence of GFP puncta indicative of interorganelle contact sites. Images labeled 1 and 2 indicate various examples of the effect that overexpression of the minimal tether had on the PM-PER split reporter. Whole-cell projections are shown. **(D)** Quantification of the effect of overexpression of the minimal tether Inp1 1–100 mRFP-Pex15 on PM-PER split reporter. Over 100 cells were analyzed for each strain. Three independent experiments were performed. Error bars represent SEM. **(E)** Inp1 1–100-mRFP-Pex15 was overexpressed under the control of the *TPI1* promoter in *rtn1/rtn2/yop1/inp1Δ* cells with Sec63-GFP and analyzed using epifluorescence microscopy. **(F)** Inp1 1–100-mRFP-Pex15 was overexpressed under the control of the *TPI1* promoter in *rtn1/rtn2/yop1/inp1Δ* cells with Sec63-GFP and BFP-PTS1 and analyzed using epifluorescence microscopy. The boxed areas are magnified. The graphs show relative fluorescence intensities of Inp1 (red), the peroxisomal matrix (blue), and the ER (green) along a line drawn through the center of the Inp1 foci. In A–C, whole-cell projections are shown. In E and F, a single focal plane is shown. Scale bars, 5 µm.

model whereby the PM-PER tether functions to ensure peroxisome retention at the cell cortex (Fig. 5 K).

It has previously been reported that peroxisomes are retained at the mother cell cortex by Inp1 acting as a "molecular hinge" and binding to both peroxisomal and ER-bound Pex3 (Knoblach and Rachubinski, 2013). However, the results obtained in our study do not support this model. Our in vivo localization studies in cER mutants consistently show Inp1 foci to be on the side of peroxisomes proximal to the plasma membrane and spatially resolved from the ER. Additionally, our Y2H study did not find any N-terminal truncations of Inp1 that interacted with Pex3. Furthermore, we were unable to reproduce specific binding of the N-terminal part of Inp1 to Pex3 under our in vitro binding assay conditions and found that only the C-terminal truncation of Inp1 (281–420) binds to Pex3 and competes with Pex19 for Pex3 binding. During the course of our work, another study shed doubt on the simplicity of the existing Pex3–Inp1–Pex3 retention model with the suggestion that additional components are required for peroxisome tethering to the cell cortex. The authors (Knoblach and Rachubinski, 2019) also suggest that contact sites between the ER and peroxisomes may occur independently of Inp1 and that Inp1 may function as a tether between peroxisomes and other organelles. Our findings are in agreement with these ideas.

Although the proposed mechanisms for the existing Pex3–Inp1–Pex3 interaction model (Knoblach and Rachubinski, 2013) do not correlate with our findings, we cannot rule out that there are additional interaction sites elsewhere in Inp1. While our minimal tether (Inp1 1–100 GFP-Pex15) and artificial tether (Num1 PH GFP-Pex15) restore peroxisome retention to the mother cell periphery in *inp1Δ* cells, peroxisome positioning is not restored to that of WT cells, with peroxisomes frequently seen "clustering" at the bud neck. This suggests that the native Inp1 tether possesses additional functionality and allows us to hypothesize that additional contacts at the cell cortex are required in order to fix peroxisomes in the correct position. As such, we do not exclude that Inp1 could tether peroxisomes via dual interaction at the cell cortex. This could be mediated by as-yet-unidentified factors that also reside at the mother cell cortex. Alternatively, this could be via other organelles such as the cER, analogous to how mitochondria are positioned by dual interaction with the plasma membrane and the cER via MECA (Lackner et al., 2013). The existence of another three-way tether involving the cER and plasma membrane would suggest there are conserved mechanisms by which organelles make contact with each other.

Inp1's role as a peroxisome tether is clear as an absence of functional Inp1 results in a complete lack of peroxisome retention in the mother cell. However, accurate peroxisome distribution at the cell periphery could also involve contact with organelles or cortical factors independent of Inp1 (Fig. 5 K). An obvious candidate protein would be Pex30, which has been shown to designate peroxisome contact sites at the ER (David et al., 2013; Mast et al., 2016). It has been reported that the absence of Pex30 and its paralogues results in increased mobility of peroxisomes, but peroxisomes are still retained at the mother cell periphery (David et al., 2013; Knoblach and Rachubinski, 2019; Munck et al., 2009). This observation could indicate a role for Pex30, or indeed other factors, in positioning and distribution of peroxisomes at the mother cell cortex which occurs in addition to the PM-PER tethering achieved by Inp1.

While the relative contributions of other cortical components in peroxisome tethering remain unclear, Inp1 is absolutely required for peroxisome retention. Our work points to a conclusion that it is Inp1 functioning as a PM-PER tether that allows Inp1 to fulfill this role. The existence of the PM-PER contact site has been previously identified (Shai et al., 2018), and we can now provide the first molecular description of a PM-PER tether and its function.

## Materials and methods

### Strains

The *S. cerevisiae* strains used in this study were derivatives of BY4741 (*MATA his3-1 leu2-0 met15-0 ura3-0*) or BY4742 (*MATα his3-1 leu2-0 lys2-0 ura3-0*) obtained from the EUROSCARF (European *Saccharomyces cerevisiae* Archive for Functional Analysis) consortium. All strains used in this study are listed in Table S1.

### Plasmids

Yeast expression plasmids were derived from Ycplac33 and Ycplac111 (Gietz and Sugino, 1988). GFP-PTS1 and mRFP-PTS1 are peroxisomal luminal marker proteins appended with peroxisomal targeting signal (Kalish et al., 1996) and were constitutively expressed under control of the *HIS3* promoter (Hoepfner et al., 2001). Yeast expression constructs used in this study were generated by homologous recombination in yeast (Uetz et al., 2000). The ORF of interest was amplified by PCR. The 5′ ends of the primers included 18-nt extensions homologous to plasmid sequences flanking the intended insertion site to enable repair of gapped plasmids by homologous recombination. The point mutant constructs were mutagenized by site-directed mutagenesis. For expression of genes under

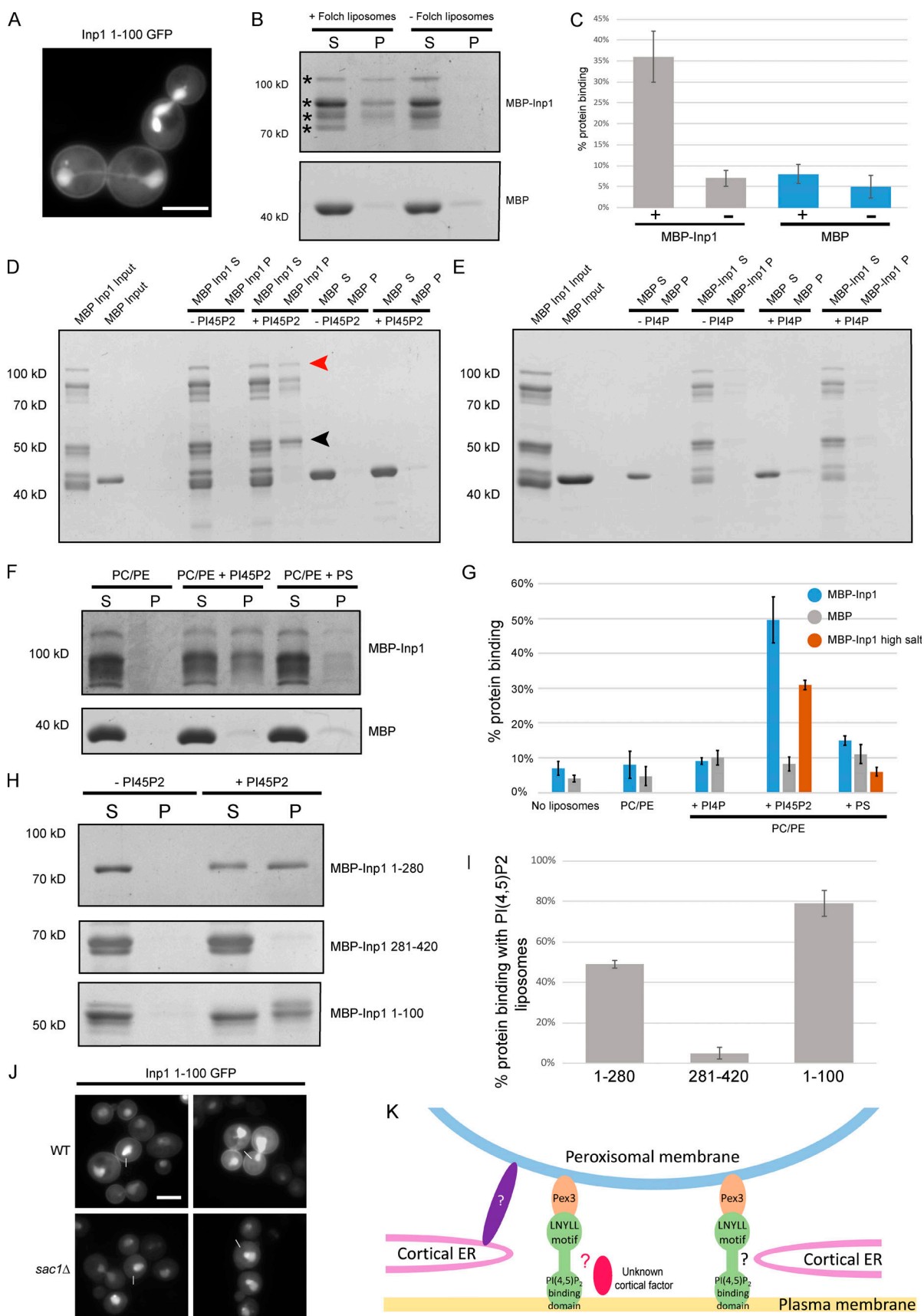

Figure 5. **Inp1 has an N-terminal PI(4,5)P₂ binding domain. (A)** An inducible truncation of 1–100 Inp1-GFP was expressed under control of the *GAL1/10* promoter in WT cells and examined by epifluorescence microscopy. A single focal plane is shown. **(B)** Purified MBP-Inp1 was incubated with Folch fraction I

liposomes and subjected to ultracentrifugation. Supernatant (S) and pellet (P) fractions were then analyzed by SDS-PAGE and Coomassie staining. Asterisks indicate multiple MBP-Inp1 fragments. **(C, G, and I)** Each supernatant and pellet band when combined were considered to represent 100% of the protein used in the experiment. + and − indicate the presence or absence of liposomes, respectively. For MBP-Inp1 high salt, purified MBP-Inp1 was incubated with liposomes made up of PC, PE with or without either $PI(4,5)P_2$ or PS in the presence of 370 mM salt (20 mM KCl + 350 mM NaCl). Liposome-binding assays were all performed in at least three independent experiments. Error bars represent SEM. **(D–F)** Purified MBP-Inp1 or MBP only were incubated with liposomes made up of PC, PE with or without 22% $PI(4,5)P_2$ (D and F), $PI(4)P$ (E), or PS (F) and subjected to ultracentrifugation. Total protein input, supernatant (S) and pellet (P) fractions were analyzed by SDS-PAGE and Coomassie staining. Red arrowhead indicates full-length MBP-Inp1. Black arrowhead indicates a breakdown product truncation of MBP-Inp1 that is still able to bind to lipids containing $PI(4,5)P_2$. **(H)** Purified truncations of MBP-Inp1 were incubated with liposomes made up of PC, PE with or without 22% $PI(4,5)P_2$ and subjected to ultracentrifugation. Supernatant (S) and pellet (P) fractions were then analyzed by SDS-PAGE and Coomassie staining. **(J)** An inducible truncation of 1–100 Inp1-GFP was expressed in WT and *sac1Δ* cells and examined by epifluorescence microscopy. White lines indicate where measurements for line-scan analyses were taken from (see Fig. S3 G). Single focal planes are shown. Multiple images from the same experiment are shown. Scale bar, 5 µm. **(K)** Models of peroxisome tethering to the mother cell cortex. Inp1 is a PM-PER tether that bridges peroxisomes and the plasma membrane by interacting with peroxisomal Pex3 via a conserved LNYLL motif and with the plasma membrane via an N-terminal domain that binds $PI(4,5)P_2$. This PM-PER tether is required for peroxisome retention. Additional peroxisome contacts with cortical structures may contribute to peroxisome positioning or distribution at the mother cell cortex. Interaction with the cER could occur via Inp1 and not-yet-identified factors (indicated by black question mark). Inp1 could also be in contact with cortical structures/factors independent of the cER (indicated by the red oval and red question mark). Peroxisome contact with the cER could also occur independently of Inp1 and via additional unidentified factors (indicated by the purple oval and white question mark).

control of their endogenous promoter, 500 or 600 nt upstream from the ORF were included. Galactose-inducible constructs contain the *GAL1/10* promoter. All yeast constructs contain the *PGK1* terminator.

For *E. coli* expression, *PEX3* (a.a.40-441) was cloned into pET30a and *INP1* into either pGEX6p-2a (Munck et al., 2009) or pMalC5X. *PEX19* was cloned into pET42a.

All plasmids generated in this study are listed in Table S2. All oligonucleotides used in this study are listed in Table S3.

### Growth conditions
Cells were grown overnight in selective glucose medium. For analysis of phenotypes by epifluorescence microscopy, cells were subsequently diluted to $OD_{600}$ = 0.1 in fresh selective glucose medium and grown for two or three cell divisions (4–6 h) before imaging. Where the induction of a protein was required (Figs. 1 A and S1), cells were grown overnight on selective raffinose medium then transferred to selective galactose medium at $OD_{600}$ = 0.1 and grown for 3–4 h.

Growth media components are as follows: minimal glucose/galactose media for selection of the uracil prototrophic marker, 2% glucose/raffinose/galactose, 0.17% yeast nitrogen base (without amino acids and ammonium sulfate), 0.5% ammonium sulfate, 1% casamino acids; minimal glucose/raffinose/galactose media for the selection of all prototrophic markers, 2% glucose/raffinose/galactose, 0.17% yeast nitrogen base (without amino acids and ammonium sulfate), and 0.5% ammonium sulfate. The appropriate amino acid stocks were added to minimal media as required.

### Mating assay
Cells of each mating type were grown to logarithmic phase on yeast peptone dextrose. Cells were then mixed, pelleted, and spotted onto a prewarmed yeast-peptone-dextrose plate and incubated at 30°C for the times indicated. Cells were harvested from the plate by scraping, resuspended into selective growth medium and imaged.

### Image acquisition
Cells were analyzed with a microscope (Axiovert 200M; Carl Zeiss) equipped with Exfo X-cite 120 excitation light source,

band-pass filters (Carl Zeiss and Chroma), and a Plan-Fluor 100×/1.45 NA or Plan-Apochromat 63× 1.4 NA objective lens (Carl Zeiss) and a digital camera (Orca ER; Hamamatsu Photonics). Image acquisition was performed using Volocity software (PerkinElmer). Fluorescence images were routinely collected as 0.5-µm z-stacks, merged into one plane in Openlab (PerkinElmer), and processed further in Photoshop (Adobe). Single focal planes are shown where indicated. Bright-field images were processed where necessary to highlight the circumference of cells in blue. For line-scan analyses, lines were drawn through a region of interest on ImageJ/FIJI to generate intensity profiles across the line for each channel.

### Cell lysis and immunoblotting
For preparation of extracts by alkaline lysis, overnight cell cultures were centrifuged and pellets resuspended in 0.2 M NaOH and 0.2% β-mercaptoethanol then left on ice for 10 min. Soluble protein was precipitated by addition of 5% TCA for a further 10 min on ice. After centrifugation (13,000 *g* for 5 min at 4°C), soluble protein was resuspended in 10 µl of 1 M Tris-HCl, pH 9.4, and boiled in 90 µl of SDS-PAGE sample loading buffer for 10 min. Samples (1–2 $OD_{600}$ equivalent) were resolved by SDS-PAGE followed by immunoblotting. Monoclonal anti-GFP antibody was obtained from Roche (11814460001). *PGK1* was used as a control (A6457; Invitrogen). Secondary antibody was HRP-linked anti–mouse polyclonal (1706516; Bio-Rad Laboratories). Monoclonal anti-polyhistidine–peroxidase was obtained from Sigma-Aldrich (A7058-1VL).

All blots were blocked in 2% (wt/vol) fat-free Marvel milk in TBS–Tween 20 (50 mM Tris-HCl, pH 7.5, 150 mM NaCl, and 0.1% [vol/vol] Tween 20). Tagged proteins were detected and imaged by chemiluminescence imaging.

### In vitro binding assays
GST-Inp1, MBP-Inp1, and His-Pex3 proteins were expressed in *E. coli* BL21 DE3. Cells were grown to $OD_{600}$ = 0.6 in 2TY media with 75 µg/ml ampicillin or 50 µg/ml kanamycin at 30°C. After 3 h of 1 mM IPTG-induced expression at 30°C, cells were harvested and the pellets resuspended in 1 ml PBS, 1 mM PMSF and EDTA-free protease inhibitor cocktail (Roche). The cells were

subjected to 1× 30-s, 2× 15-s sonication at an amplitude of 12 μm and samples were kept on ice throughout. 1% Triton X-100 was added to the lysates (except when lysate contained Pex3) and incubated at 4°C for 30 min. Lysates were subsequently centrifuged at 20,000 $g$ for 5 min and the supernatant retained. Glutathione Sepharose 4B (GE Healthcare), amylose (New England Biolabs), or Ni-Sepfast (BioToolomics) beads prewashed in PBS were added to fusion protein lysates, respectively, and incubated at 4°C for 30 min. The beads were washed twice with 1× PBS + 0.05% Igepal CA-630 (Sigma-Aldrich) before relevant secondary *E. coli* lysates were added. Samples were then incubated for 2 h at 4°C with end-over-end mixing. The beads were washed four times with 1× PBS + 0.05% Igepal CA-630. The second wash included 500 mM KCl, as this was found to be sufficient to remove any nonspecific binding to bead only controls without affecting binding, as previously described (Munck et al., 2009). Bound material was eluted with SDS-PAGE sample loading buffer, resolved by SDS-PAGE, and analyzed by Coomassie staining.

### In vitro competition assay
MBP-Inp1, GST-Pex19, and His-Pex3 proteins were expressed in *E. coli* BL21 DE3. Cells were grown and induced and lysates prepared as above. Total lysates containing GST-Pex19 were incubated with washed glutathione Sepharose 4B beads at 4°C for 30 min. Total lysates containing His-Pex3 and MBP-Inp1 proteins were mixed together and incubated at 4°C for 30 min. The glutathione Sepharose beads were washed twice with 1× PBS + 0.05% Igepal CA-630 before the His-Pex3/MBP-Inp1 lysate mixtures were added. Samples were then incubated for 2 h at 4°C with end over end mixing. The total unbound fraction was removed from the glutathione Sepharose beads and added immediately to amylose resin beads. These samples were then incubated for a further 1 h at 4°C with end-over-end mixing. All beads were washed four times with 1× PBS + 0.05% Igepal CA-630, with the second wash containing 500 mM KCl (as described above). Bound fractions were eluted from the glutathione Sepharose beads using SDS-PAGE sample loading buffer. Bound fractions were eluted from amylose beads with maltose buffer (20 mM Tris-HCl, pH 7.4, 200 mM NaCl, 1 mM EDTA, and 10 mM maltose). All samples were resolved by SDS-PAGE and analyzed by Coomassie staining. For detection of His-Pex3–MBP-Inp1 complexes, immunoblot analysis was performed with the monoclonal anti-polyhistidine peroxidase antibody (A7058-1VL; Sigma-Aldrich).

### Preparation of liposomes and lipid-binding assay
For preparation of Folch liposomes, 22 μl of a 25 mg/ml solution of Folch fraction-1 (Sigma-Aldrich) was dried under nitrogen and then resuspended in 200 μl liposome buffer (20 mM Hepes, pH 7.2, 100 mM KCl, 2 mM MgCl$_2$, and 1 mM DTT) at 60°C for 30 min with gentle agitation. For preparation of synthetic liposome solutions, 40 μl of 25 mg/ml PC solution, 4.6 μl of 25 mg/ml PE solution, and 307 μl of 1 mg/ml PI(4,5)P$_2$ or PI(4)P solution or 12.3 μl 25 mg/ml PS solution (dissolved in chloroform; Avanti Polar Lipids) were dried under nitrogen then resuspended in 400 μl liposome buffer. MBP-Inp1 was eluted from amylose beads in maltose buffer (20 mM Tris-HCl, pH 7.4, 200 mM NaCl, 1 mM EDTA, and 10 mM maltose). The purified protein was prespun at 350,000 $g$ for 15 min (Ultra

centrifuge, TL100 rotor; Beckman) then immediately added to 10 μl of liposome solution and made up to final volume 50 μl with maltose buffer. The liposome–protein mixture was incubated at room temperature for 30 min before pelleting the liposomes at 280,000 $g$ for 15 min. After centrifugation, supernatants and pellets were separated and pellets resuspended in 50 μl liposome buffer. 10 μl Strataclean resin (Stratagene) was added to each sample to concentrate the protein. This was pelleted by spinning at 8,000 $g$ in a table-top microcentrifuge for 2 min. Supernatants were removed and the pellet of each sample was resuspended in 15 μl SDS-PAGE loading buffer. Samples were then analyzed by SDS-PAGE and Coomassie staining.

### Online supplemental material
Fig. S1 provides additional evidence to support the data shown in Fig. 1 regarding the importance of the LXXLL motif in both Inp1 and ScPex19. Fig. S2 supports the data shown in Fig. 2 and provides additional evidence that MBP-Inp1 C outcompetes GST-Pex19 for binding to His-Pex3 in vitro. Fig. S3 supports the data shown in Figs. 4 and 5 and provides evidence that overexpression of the minimal tether does not affect peroxisome biogenesis. Table S1 lists the strains used in this study. Table S2 lists the plasmids used in this study. Table S3 lists the oligonucelotides used in this study.

## Acknowledgments
We would like to thank William Prinz (National Institute of Diabetes and Digestive and Kidney Diseases, Bethesda, MD) for the *rtn1/rtn2/yop1Δ* strain and Scott Emr (Cornell University, Ithaca, NY) for the *scs2/scs22/ist2/tcb1/tcb2/tcb3Δ* strain. We would also like to thank Isobel Gibson (University of Sheffield, Sheffield, UK) for construction of the Inp1 1-100 GFP-Pex15 plasmid, as well as Richard Rachubinski and Ida van der Klei for sharing unpublished data.

The authors declare no competing financial interests.

Author contributions: G.E. Hulmes performed most experiments. J.D. Hutchinson and J.M. Nuttall performed experiments to provide preliminary data. E.G. Allwood provided technical insight and reagents for liposome binding assays. G.E. Hulmes, J.D. Hutchinson, N. Dahan, J.M. Nuttall, E.G. Allwood, K.R. Ayscough, and E.H. Hettema analyzed data. G.E. Hulmes and E.H. Hettema wrote the manuscript. All authors provided feedback on the manuscript.

Submitted: 4 June 2019

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

# Supplemental material

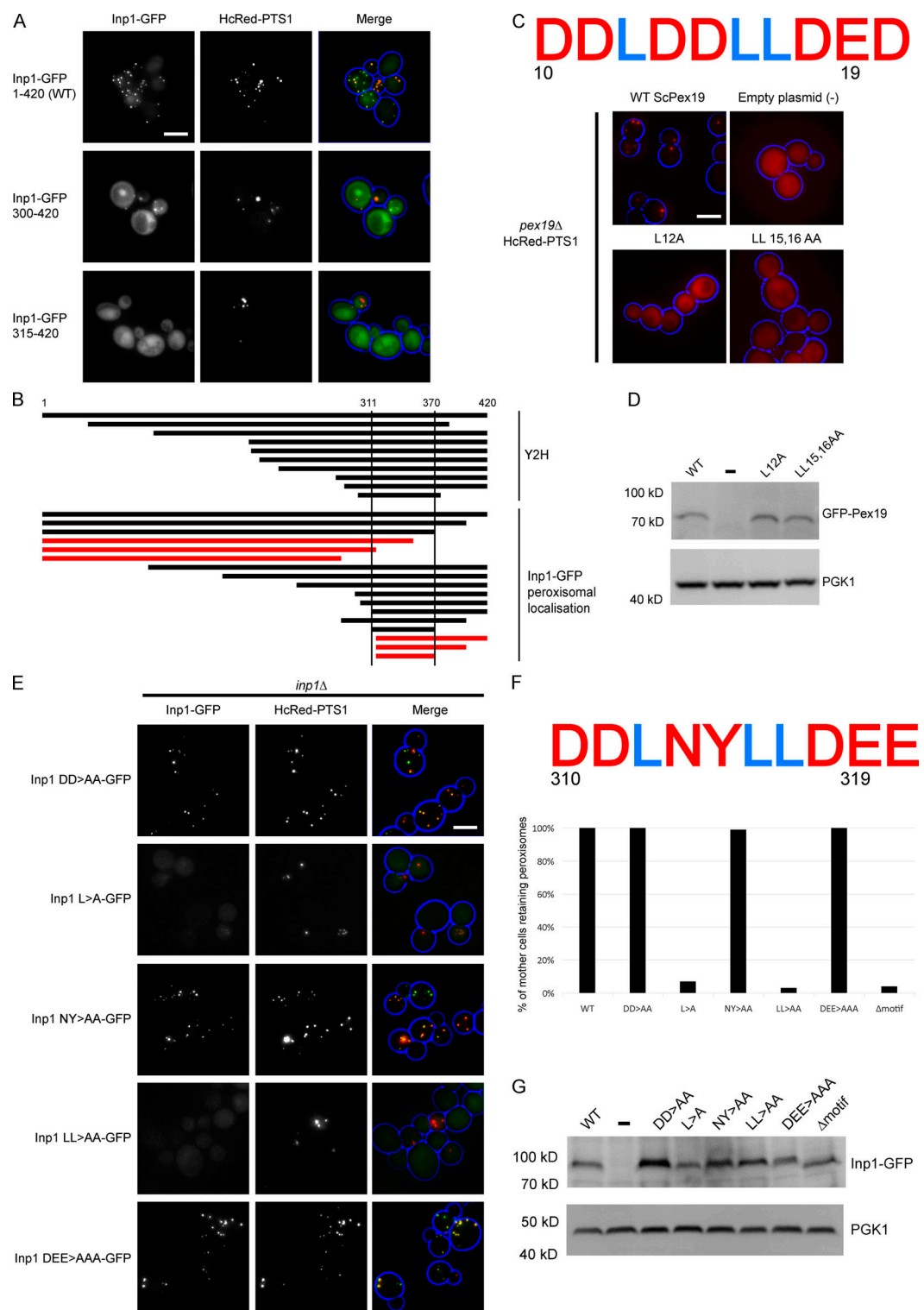

Figure S1.   **Conserved LXXLL motifs in both Inp1 and Pex19 affect their respective functions. (A)** Truncations of Inp1-GFP under control of the *GAL1/10* promoter were coexpressed in *inp1Δ* cells with constitutively expressed HcRed-PTS1 and examined by epifluorescence microcopy. **(B)** A diagram showing the truncations of Inp1 identified in a genome-wide yeast two-hybrid screen with Pex3 as bait (upper panel) or that localize to peroxisomes (black lines; lower panel). The truncations represented with red lines did not colocalize with peroxisomes. **(C)** GFP-ScPex19 leucine motif mutants were expressed under control of the *PEX19* promoter in *pex19Δ* cells expressing HcRed-PTS1. (-), an empty plasmid control. The motif region sequence (amino acids 10–19) is shown. **(D)** Immunoblot analysis of GFP-Pex19 mutants. (-), empty plasmid control. *PGK1*, loading control. **(E)** Alanine scan of Inp1-GFP region 310–319. Mutants were expressed under control of the *INP1* promoter in *inp1Δ* cells coexpressing HcRed-PTS1. Scale bar, 5 µm. **(F)** Quantitative analysis of peroxisome retention in alanine scan mutants. Budding cells were scored for the presence or absence of peroxisomes in the mother cell. At least 100 budded cells were quantified for each mutant. The motif region sequence targeted for mutagenesis (amino acids 310–319) is shown. **(G)** Immunoblot analysis of Inp1-GFP mutants. (-), empty plasmid control. *PGK1*, loading control.

Hulmes et al.
The Pex3–Inp1 complex tethers yeast peroxisomes to the plasma membrane

**Journal of Cell Biology**   S2

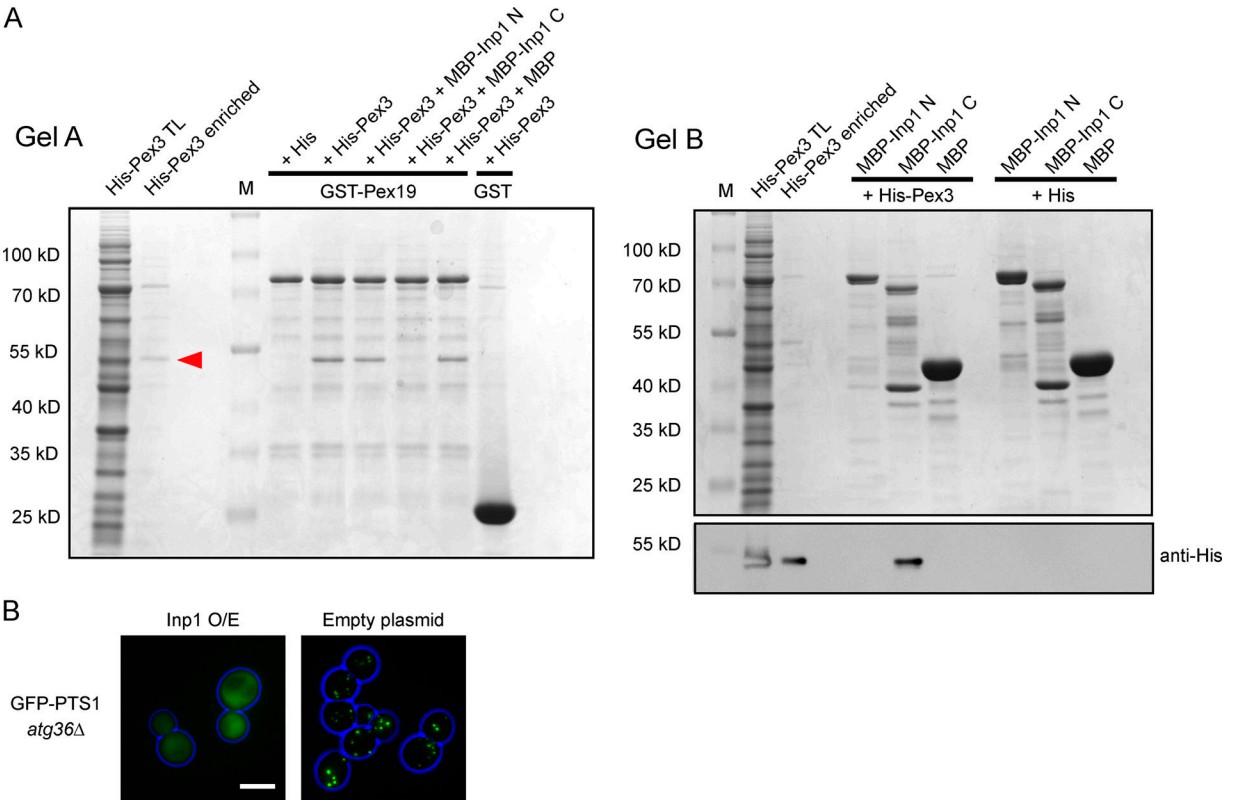

**Figure S2.** **Inp1 competes with Pex19 in vitro for Pex3 binding. (A)** *E. coli* total lysates expressing His-Pex3 (40–441) or His-tag only were mixed with *E. coli* total lysates containing MBP-Inp1 N (1–280), MBP-Inp1 C (281–420), or MBP. These lysate mixtures were then incubated with GST-Pex19 or GST and immobilized on glutathione Sepharose beads. After this incubation, the unbound fraction was removed and incubated with amylose beads. The glutathione Sepharose (gel A) and amylose beads (gel B) were washed extensively. Protein was eluted from glutathione Sepharose beads with protein loading buffer, and bound fractions were analyzed by SDS-PAGE and Coomassie staining of the gel (gel A). A lane was included with enriched His-Pex3 as control. MBP-Inp1 C prevents binding of His-Pex3 to GST-Pex19. Protein was eluted from the amylose beads in 10 mM maltose buffer, and bound fractions were analyzed by SDS-PAGE and Coomassie staining of the gel (gel B). Because the bound His-Pex3 is not clearly distinguishable by Coomassie stain on gel B as a consequence of MBP-Inp1 breakdown bands, a Western blot with anti-His was done to show the presence of His-Pex3 bound to MBP-Inp1 C. M, molecular weight marker. Red arrow indicates His-Pex3. Protein visible on Gel B is 1/4 of total protein incubated with GST-Pex19. Protein visible on anti-His blot is 1/10. **(B)** *atg36Δ* cells were transformed with GFP-PTS1 and Inp1 under the control of the *TPI1* promoter or an empty plasmid. The cells were then examined by epifluorescence microscopy. Scale bar, 5 µm.

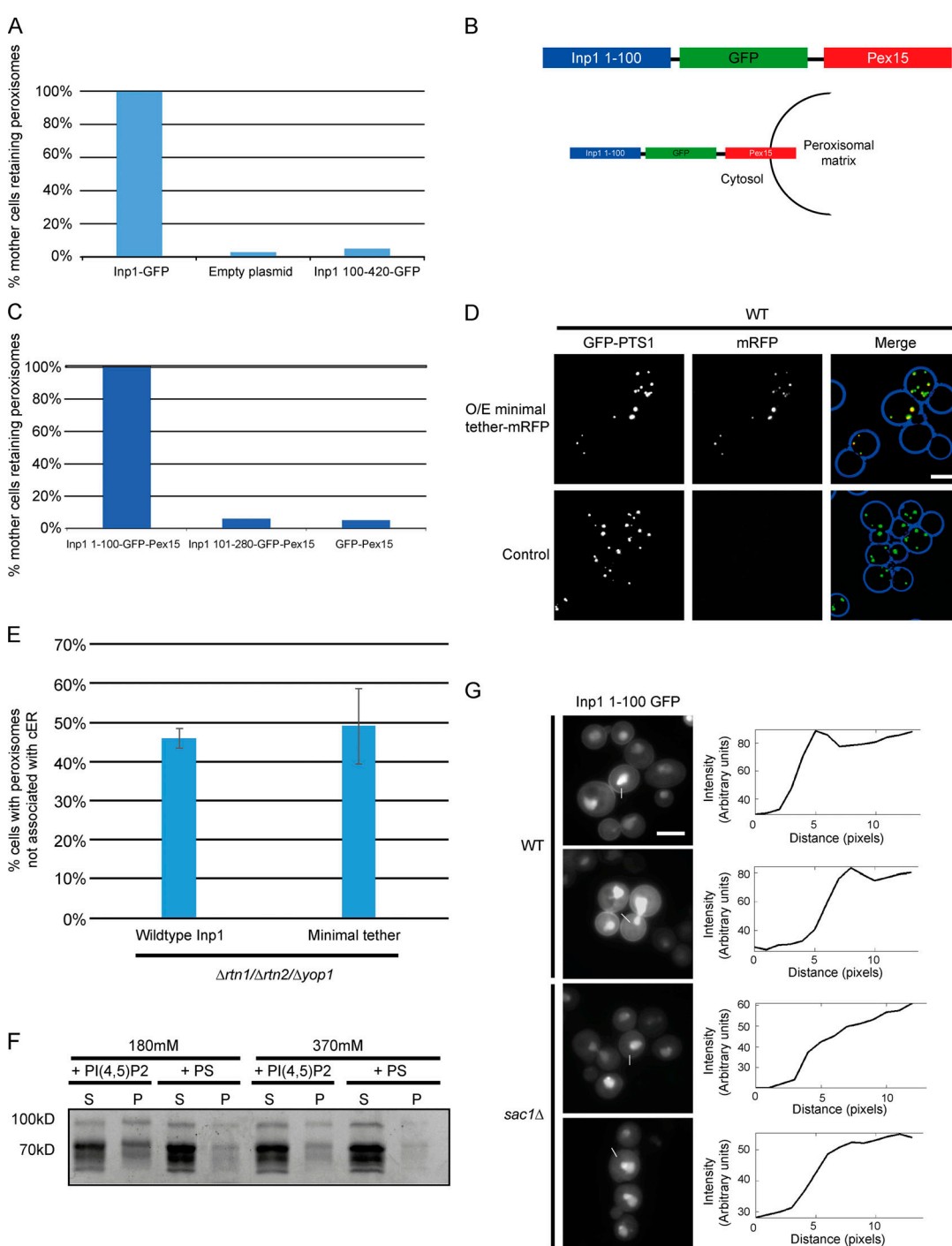

Figure S3. **Inp1 meets the criteria to be classified as a PM-PER tether. (A)** Quantitative analysis of the assay described in Fig. 4 A. Mating cell pairs were scored for the presence of peroxisomes in the mother cell. At least 100 budding cells were quantified for each strain. **(B)** A schematic diagram illustrating how the Inp1 component of the minimal tether fusion protein described in Fig. 4 B is present on the cytosolic face of the peroxisomal membrane. **(C)** Quantitative analysis of the assay described in Fig. 4 B. Mating cell pairs were scored for the presence of peroxisomes in the mother cell. At least 100 budding cells were quantified for each strain. **(D)** The minimal Inp1 tether 1–100 mRFP-Pex15 was overexpressed (O/E) under control of the TPI1 promoter in WT cells with the constitutively expressed peroxisomal matrix marker GFP-PTS1. Cells were analyzed with epifluorescence microscopy. Whole-cell projections are shown. **(E)** Quantitative analysis of the assay described in Fig. 4 F. Budding *rtn1/rtn2/yop1Δ* cells were scored for the presence of peroxisomes at the periphery but not associated with visible ER. Over 100 budding cells were quantified for each strain. Three independent experiments were performed. Error bars represent SEM. **(F)** Purified MBP-Inp1 was incubated with liposomes made up of PC, PE with or without either PI(4,5)$P_2$ or PS in the presence of 180 mM salt (20 mM KCl + 160 mM NaCl) or 370 mM salt (20 mM KCl + 350 mM NaCl). Samples were then subjected to ultracentrifugation. Supernatant (S) and pellet (P) fractions were analyzed by SDS-PAGE and Coomassie staining. **(G)** An inducible truncation of 1–100 Inp1-GFP was expressed in WT and *sac1Δ* cells and examined by epifluorescence microscopy. These images are the same as those shown in Fig. 5 J. The graphs show relative fluorescence intensities of Inp1 1–100 GFP across the plasma membranes. White lines indicate where the measurements were taken from. Single focal planes are shown. Multiple images from the same experiment are shown. Scale bars, 5 µm.

**Provided online are three tables. Table S1 lists the strains used in this study. Table S2 lists the plasmids used in this study. Table S3 lists the oligonucleotides used in this study.**

