## [Peer Review File · The Journal of Cell Biology]

The Pex3/Inp1 complex tethers yeast peroxisomes to the plasma membrane

Georgia Hulmes, John Hutchinson, Noa Dahan, James Nuttall, Ellen Allwood, Kathryn Ayscough, and Ewald Hettema

Corresponding Author(s): Ewald Hettema, University of Sheffield

Review Timeline:

Submission Date:	2019-06-04
Editorial Decision:	2019-08-07
Revision Received:	2020-03-09
Editorial Decision:	2020-03-29
Revision Received:	2020-04-26
Accepted:	2020-06-18

Monitoring Editor: Jodi Nunnari

Scientific Editor: Marie Anne O'Donnell

Transaction Report:

DOI: <https://doi.org/10.1083/jcb.201906021>

July 29, 2019

Re: JCB manuscript #201906021

Dr. Ewald H Hettema
University of Sheffield
Department of Molecular Biology and Biotechnology Firth Court
Western Bank
Sheffield S10 2TN
United Kingdom

Dear Dr. Hettema,

Thank you for submitting your manuscript entitled "The Pex3/Inp1 complex tethers yeast peroxisomes to the plasma membrane". The manuscript has been evaluated by expert reviewers, whose reports are appended below. Unfortunately, after an assessment of the reviewer feedback, our editorial decision is against publication in JCB.

You will see that all the reviewers note that the identification of Pex3-Inp1 as the first tether of peroxisomes to the plasma membrane is novel and interesting but they are quite critical of the overall advance as Pex3-Inp1 has already been proposed to control peroxisomal inheritance via their tethering to the ER. To be suitable for JCB, we feel that some expansion to the scope would be necessary, such as providing more insight into the relationship between ER and plasma membrane tethering by Inp1 in this process, how tethering to the plasma membrane specifically alters peroxisomal retention, or identifying other peroxisome processes that may require tethering to the plasma membrane via Pex3-Inp1, or the role of Pex3 versus Pex19 binding in the tethering. In addition, the reviewers note that further evidence is necessary to bolster the main claims and confirm that Inp1 recognizes PI(4,5)P2 in the plasma membrane and clarify if this alone, or in conjunction with ER tethering, is necessary for Inp1 to mediate peroxisomal inheritance.

Although your manuscript is intriguing, I feel that the points raised by the reviewers are more substantial than can be addressed in a typical revision period. If you wish to expedite publication of the current data, it may be best to pursue publication at another journal.

Given interest in the topic, I would be open to resubmission to JCB of a significantly revised and extended manuscript that fully addresses the reviewers' concerns and is subject to further peer-review. If you would like to resubmit this work to JCB, please contact the journal office to discuss an appeal of this decision or you may submit an appeal directly through our manuscript submission system. Please note that priority and novelty would be reassessed at resubmission.

Regardless of how you choose to proceed, we hope that the comments below will prove constructive as your work progresses. We would be happy to discuss the reviewer comments further once you've had a chance to consider the points raised in this letter. You can contact the journal office with any questions, cellbio@rockefeller.edu.

As an alternative to expanding the scope of the study for JCB, we have discussed your manuscript with the editors of Life Science Alliance (<http://www.life-science-alliance.org/>) and they would

consider a revision that provides: a pbp response and accordingly text changes, as well as addressing the request for controls (rev#1 and #2), quantifications (rev#1+ #3) and statistics (rev#3), addressing the specific comments of rev#3 (the comment regarding the competition binding assay does not need to get addressed). Following the reviewer input on how to dissect the relative contribution of PM binding to PO retention (rev#3) would strengthen the paper significantly and elevate its value to the community. LSA is our academic editor-led, open access journal launched as a collaboration between RUP, EMBO Press and Cold Spring Harbor Press. You can use the link below to initiate an immediate transfer of your manuscript files and reviewer comments to LSA.

Link Not Available

Thank you for thinking of JCB as an appropriate place to publish your work.

Sincerely,

Jodi Nunnari, Ph.D.
Editor-in-Chief

Marie Anne O'Donnell, Ph.D.
Scientific Editor

Journal of Cell Biology

Reviewer #1 (Comments to the Authors (Required)):

Hulmes et al identify Inp1 as a peroxisome-plasma membrane tether in the budding yeast *S. cerevisiae*. Previous work has shown that Inp1 functions to tether peroxisomes to the ER. Here the authors provide compelling evidence to suggest that Inp1 tethers peroxisomes directly to the plasma membrane. Through in vivo and in vitro structure-function studies the authors identify a Pex3 binding motif on Inp1 that is required for recruitment to peroxisomes and a region in the N-terminus of the protein that is required for peroxisome retention and binding to the plasma membrane, likely via an interaction with PI(4,5)P₂. Using a reticulon mutant, the authors are able to demonstrate that Inp1 can tether peroxisomes to the plasma membrane independently of the ER. The paper is well written and provides the first molecular description of a peroxisome-plasma membrane tether.

There are a few concerns that should be addresses prior to publication.

Major comments:

1. To strengthen the conclusion that Inp1 binds PI(4,5)P₂ in the plasma membrane (PM), the authors should do the following:

a. To appropriately control the in vitro liposome binding experiments and rule out that the interaction with PI(4,5)P₂ is not simply due to a charge effect, i.e. due to the fact that PI(4,5)P₂ has a greater negative charge than PI4P, the negative charge on the control lipids needs to be equivalent to that of the PI(4,5)P₂-containing lipids. PS can be added to the liposomes to balance

the charge or PI(3,5)P2 could be used, which would have an equivalent charge to PI(4,5)P2.

b. The addition of the *sac1* mutant data nicely supports the conclusion that Inp1 interacts with PI(4,5)P2. To strengthen the *sac1* mutant data further, the authors should in some way quantify that the enrichment of Inp1 1-100 GFP along the PM is reduced in the absence of Sac1. Perhaps a quantitative line scan analysis can be used to show an enrichment of fluorescent signal at the PM in WT cells that is absent in the *sac1* mutant.

2. The idea that Inp1 can function as a peroxisome-PM tether is a major conclusion of the paper. Thus, the authors should strengthen the ER data shown in Fig. 5A and B by providing quantification. Once again, perhaps line scan analysis can be used to clearly show Inp1 foci can be found on ER-free regions of the cell cortex.

3. The authors should include a discussion of how they think the peroxisome-PM tethering function of Inp1 relates to its peroxisome-ER tethering function; are they mutually exclusive, can Inp1 function in both tethering roles simultaneously? Such discussion would be greatly strengthened by the addition of experimental evidence that can be used to determine if one or both of the tethering functions of Inp1 is required for peroxisome retention.

Minor comments:

1. It would be helpful if amino acid numbers were added to the Inp1 schematic in Fig. 1C.

2. For the cell images shown, it should be stated if they are maximum intensity projects or single focal planes.

3. The legend for Fig. 3A and B states "mating pairs were scored..." but I believe it should be budding cells were scored.

Reviewer #2 (Comments to the Authors (Required)):

Inp1 and Pex3 are necessary for proper peroxisome inheritance (Knoblach et al, EMBO 2013). This study suggests that Inp1 actually binds the plasma membrane and anchors peroxisomes to the plasma membrane by interacting with Pex3, forming plasma membrane-peroxisome contact sites. It also characterizes the Inp1 binding by Pex3 and show that they interact via a motif that is similar to the one in Pex19 that is bound by Pex3. The work is well done and largely convincing, except for the claim that Inp1 specifically binds PI(4,5)P2. However, even if more convincing evidence were presented, the study is only a modest advance. It is certainly interesting that yeast forms ER-plasma membrane contact sites, but it was already known that the Pex3-Inp1 tether plays a role in peroxisome inheritance. What is the significance of the fact that the Inp1-Pex3 tether anchors peroxisomes to the plasma membrane rather than the ER? Perhaps the most interesting aspect of this study is the suggestion that peroxisomes may form three-way contacts with the ER and plasma membrane (as Lacker et al showed for mitochondria). However, it does not definitively show this let alone address the importance of the three-way contacts. There are two ways the study could be improved.

1. The liposome binding studies lack controls to show that the binding is specific and is not simply driven by charge. Other PIPs or lipids with equivalent charge should be used.

2. The competition of Inp1 and Pex19 for Pex3 binding is interesting, but it is not clear that it has any functional relevance. Does Pex19 binding to Pex3 regulate contact site formation?

Reviewer #3 (Comments to the Authors (Required)):

Summary:

Hulmes et al. have identified and characterised the Pex3/Inp1 complex as the first known peroxisome (PO)-plasma membrane (PM) tether in *S. cerevisiae*, and determined that this tethering is required for PO positioning and retention in the mother cell during cell division. Through expressing truncations/mutants in Δ inp1 cells, a yeast 2-hybrid screen and in vitro binding assays, they have identified the C-terminal motif of Inp1 that is required for its PO localisation via a direct interaction with Pex3, which in turn is required for PO retention during budding. By expressing an artificially PO-tethered Inp1 truncation, they have demonstrated that the N-terminus is necessary and sufficient for PO retention. The overexpressed N-terminus of Inp1 localises to the cell periphery in a PI(4,5)P₂-dependent manner, and binds specifically to PI(4,5)P₂ in liposome binding assays. They show that Inp1 possesses the necessary characteristics to suggest it is a bona-fide tether component by expressing an artificial tether to rescue the Δ inp1 phenotype, and showing that overexpression of this tether increases the number of PO-PM contacts. Altogether, this demonstrates that Inp1 can tether peroxisomal Pex3 and PM PI(4,5)P₂ to mediate PO retention during cell division.

General comments:

Even though the existence of a PO-PM tether has already been observed in yeast, both the characterisation of its molecular components and its function in PO retention described in this study are novel and interesting. However, the Pex3-Inp1 interaction has already been characterised in *S. cerevisiae* with respect to forming PO-ER tethers, and has been shown to regulate PO retention during cell division. I would like to see more evidence and discussion pertaining to Inp1's alternative role as a PO-ER tether, which is only briefly covered. Are the authors disputing the current model that Inp1-dependent PO-ER contacts mediate PO retention, or suggesting a parallel/additional pathway exists, or suggesting that both contacts acting together are important for retention? I think it would be necessary to resolve this, and determine whether PO-PM contacts, as opposed to/as well as PO-ER contacts, play a significant physiological role in PO retention.

Overall, the data presented are convincing, thorough and well-controlled, and the conclusions appropriate, with the authors using a combination of in vitro and cellular assays to support their claims.

Specific comments:

Introduction:

- An introduction to POs might be helpful
- It would be helpful to note here that Pex3 is a peroxisomal membrane protein (PMP) itself

Results and discussion:

- In Figure 1A, it is not shown that the Inp1 puncta necessary correspond to PO. This is, however, shown in Figure S1A - this is important so it might be clearer if this was moved to the main Figure?
- Western blots for the various protein tags are not consistently shown for the in vitro binding experiments. Although the experiments are well-controlled, blots would be useful to confirm that the bands seen on the Coomassie gels are what they are identified as.

- In the competition binding assay, the authors show that only the C terminal of Inp1 can successfully compete for Pex3 binding with Pex19. Since the authors identify the LXXLL motif as the binding site, does a recombinant version of their LL>AA Inp1 mutant fail to compete with Pex19? Do the authors consider that this Inp1/Pex19 competition might be physiologically relevant?
- In the section on Inp1/Pex19 competition for Pex3 binding, the rationale for using the $\Delta mdh3/\Delta gpd1$ mutant is not clear - the experiment needs to be explained better. What does this reveal about Inp1 function?
- The authors state that: "The absence of PO in cells overexpressing Inp1 is initially a result of over-retention in mother cells with buds failing to inherit POs". Do they have any evidence to support this e.g. a time-course during division, or is there a suitable reference?
- In Figure 3A, the authors say they observe PO retention in the mother cells. How are the mother and daughter cells distinguished? In addition, this panel shows a greater number of PTS1 puncta (= PO?) in cells in which POs are retained - why does this occur? This needs to be explained as it confounds interpretation.
- The authors provide a hypothesis for the different PO clustering seen when the minimal Inp1 tether is expressed, but this is somewhat confusing and needs to be explained better (there is a clearer explanation later in the manuscript).
- The experiments shown in Fig 4 are very nice, but the interaction of MBP-Inp1 with PI(4,5)P2 is not quantified, unlike the others - this should be added.
- The experiments showing Inp1 and PO localisation at the PM are probably the weakest, as a result of the inherent confounding factor of cortical ER positioning. The authors have come up with the creative solution of using a mutant lacking cortical ER tubules, but it would be necessary to show if PO retention is normal in the mutant for this to be a suitable model system. This would perhaps also inform the ER-PO vs PM-PO issue. Also, only one example is shown (Fig. 5B) of Inp1 localising to the side of the PO next to the periphery and away from the ER - how typical was this localisation? The authors might also like to consider using a technique with sufficient resolution to identify PO-PM contacts in WT cells e.g. EM to demonstrate that these contacts form under physiological conditions.
- In Figure 5B, the co-localised puncta of ER, PO and Inp1 are hard to see - enlarged ROIs would make this clearer. Does the inclusion of 3 z planes add anything - would the middle image not suffice?
- Are the total number of POs the same in control and minimal-tether-overexpressing cells (Figure 5G)? If not, this could confound the observation that expression of the tether increases the number of PO-PM contacts.
- Fig. 5G, PM-PER - should the PO not be associated with the PM more strongly (image 1) under these conditions?
- The presentation of the data indicating that overexpression of the minimal tether increases the number of PM-PER puncta per cell (Figure 5H) is a little confusing and could be made clearer.
- Would it be possible to design minimal tethers, or use mutant strains to separate Inp1's ER and PM binding functions, e.g. express Inp1 in which only PI(4,5)P2 binding was disrupted? This would allow the relative contribution of each to a) contact site number and b) PO retention to be determined.
- The authors end by proposing the hypothesis that Inp1 tethers PO via dual interactions with the ER and the PM (although they have not addressed Inp1-dependent PO-ER contacts anywhere in their data). Do they imagine that peroxisomes could interact with both ER and PM at the same time? Can they speculate why such three-way positioning might be important? Contacts between one organelle with multiple others are an upcoming hot topic so some discussion here would be very interesting.

Throughout:

- No statistical analysis is presented anywhere and should be included. It is sometimes unclear how many independent experiments were performed.

RE manuscript #201906021

Dear editors,

As requested, please find attached our detailed response to the decision letter and the comments of the reviewers. As you will see, we have significantly revised and extended our manuscript and fully addressed the reviewers concerns.

We are awaiting your advice as to whether we resubmit our manuscript or whether we should submit it as a new manuscript.

First, the authors would like to thank the editor and reviewers for their critical input. The extensive comments from the reviewers were extremely helpful and we have now been able to reach a firm conclusion that Inp1-Pex3 complex tethers peroxisomes to the plasma membrane and that its plasma membrane binding is required for peroxisome retention in the mother cell during cell division. We also show that retention of peroxisomes can occur independent of interaction with the cortical ER and present a new model for peroxisome retention.

Firstly, in response to the comments in the editor's decision letter:

- 1) You will see that all the reviewers note that the identification of Pex3-Inp1 as the first tether of peroxisomes to the plasma membrane is novel and interesting but they are quite critical of the overall advance as Pex3-Inp1 has already been proposed to control peroxisomal inheritance via their tethering to the ER. To be suitable for JCB, we feel that some expansion to the scope would be necessary, such as providing more insight into the relationship between ER and plasma membrane tethering by Inp1 in this process, how tethering to the plasma membrane specifically alters peroxisomal retention, or identifying other peroxisome processes that may require tethering to the plasma membrane via Pex3-Inp1, or the role of Pex3 versus Pex19 binding in the tethering.***

To expand the scope of our manuscript and "provide more insight into the relationship between the ER and plasma membrane tethering" we have tested the contribution of the ER in peroxisome retention in more detail, using two mutants that lack typical cortical ER structures. Both mutants retain and segregate peroxisomes properly during cell division and we find peroxisomes are still present at the cell cortex, away from the collapsed or disrupted ER. We solidify our narrative that Inp1 meets all criteria for being a plasma membrane-peroxisome tether by showing more clearly and with quantification that in disrupted ER mutants, Inp1 can be seen juxtaposed to peroxisomal foci, more proximal to the plasma membrane and spatially resolved from the ER. These are very informative experiments as they clearly show that peroxisomes can be retained by binding the plasma membrane and that this process does not require the cortical ER.

In terms of "how tethering to the plasma membrane specifically alters peroxisomal retention," we find that binding to the plasma membrane is required for peroxisome retention. We further show that Inp1 contains a PM binding domain and that deletion of this domain stops peroxisome retention. Additionally, the plasma membrane binding domain of Num1 can act as an artificial tether in the absence of Inp1, to restore retention of peroxisomes to the cell periphery. The PM binding domains of Inp1 and Num1 fused to a tail anchored peroxisomal protein restore cortical retention, however, distribution along the cortex is affected. This suggests that additional cortical factors are required for positioning peroxisomes and that this is dependent on Inp1. We discuss this.

We are now able to be more explicit in pointing out that we can no longer support the current Pex3-Inp1-Pex3 model of peroxisome retention in mother cells in which Inp1 was proposed to be a molecular hinge between peroxisomal and ER bound Pex3 (Knoblach et al., 2013). Interestingly, doubt has recently been shed onto their model by the authors themselves and they propose in their most recent study (Knoblach and Rachubinski, 2019) that contact sites between the ER and peroxisomes may occur independently of Inp1 and that Inp1 may function as a tether between peroxisomes and other organelles. Our findings are in full agreement with these ideas.

2) *In addition, the reviewers note that further evidence is necessary to bolster the main claims and confirm that Inp1 recognizes PI(4,5)P₂ in the plasma membrane and clarify if this alone, or in conjunction with ER tethering, is necessary for Inp1 to mediate peroxisomal inheritance.*

We have furthered our evidence that Inp1 binds PI(4,5)P₂ preferentially over PS and PI(4)P in our *in vitro* liposome binding assays.

We now clarify that PM binding is necessary for peroxisome retention and that this can take place in the absence of cortical ER. Additionally, an artificial peroxisome anchored tether consisting of a lipid binding domain only e.g. the first 100 amino acids of Inp1 or the PH domain of Num1 can restore peroxisome retention to the mother cell cortex.

To emphasise our point, our improved manuscript now shows the localisation of the minimal Inp1 plasma membrane binding tether in a disrupted cortical ER mutant. We see that the minimal Inp1 tether partially overlaps with peroxisomal foci, on the side more proximal to the plasma membrane and spatially resolved from cortical ER. This tether is seen to attach peroxisomes to the mother cell periphery even in areas where there is no cortical ER. This allows us to clarify that PM binding alone is sufficient for peroxisome retention and enforces our conclusion that peroxisome-ER contact is not essential for retention.

We have also addressed the reviewers comments in full, please find below our formal responses to the reviewers comments:

Reviewer #1:

The authors would like to thank this reviewer for their constructive comments. We have implemented all of them in our improved manuscript.

1. To strengthen the conclusion that *Inp1* binds *PI(4,5)P2* in the plasma membrane (PM), the authors should do the following:

a. To appropriately control the *in vitro* liposome binding experiments and rule out that the interaction with *PI(4,5)P2* is not simply due to a charge effect, i.e. due to the fact that *PI(4,5)P2* has a greater negative charge than *PI4P*, the negative charge on the control lipids needs to be equivalent to that of the *PI(4,5)P2*-containing lipids. PS can be added to the liposomes to balance the charge or *PI(3,5)P2* could be used, which would have an equivalent charge to *PI(4,5)P2*.

We have now carried out assays with liposomes containing PS to rule out that *PI(4,5)P₂* interaction is not due to a negative charge effect (Fig. 5F, G). As the reviewers will see, MBP-*Inp1* does not interact with PC/PE liposomes supplemented with the negative charged phospholipids *PI(4)P* or PS but does bind in the presence of *PI(4,5)P₂*.

b. The addition of the *sac1* mutant data nicely supports the conclusion that *Inp1* interacts with *PI(4,5)P2*. To strengthen the *sac1* mutant data further, the authors should in some way quantify that the enrichment of *Inp1* 1-100 GFP along the PM is reduced in the absence of *Sac1*. Perhaps a quantitative line scan analysis can be used to show an enrichment of fluorescent signal at the PM in WT cells that is absent in the *sac1* mutant.

Quantitative line scan analyses have been now carried out to further visualise the enrichment of the fluorescent signal at the plasma membrane in WT cells compared to the *sac1Δ* mutant (Fig. S3F).

2. The idea that *Inp1* can function as a peroxisome-PM tether is a major conclusion of the paper. Thus, the authors should strengthen the ER data shown in Fig. 5A and B by providing quantification. Once again, perhaps line

scan analysis can be used to clearly show Inp1 foci can be found on ER-free regions of the cell cortex.

As you point out, Inp1 functioning as a PM-PER tether is a major conclusion and as such we have now rewritten the manuscript to address this point first. We have now quantified the distribution of peroxisomes in the ER mutants and also the number of cells with peroxisomes present at the periphery of ER mutants in the absence of visible ER. (Fig. 1B, E). Line scan analyses have been carried out which show the location of peroxisomes at the plasma membrane and which clearly show that Inp1 foci are seen on peroxisomes at the cell cortex free of or spatially resolved from the ER (Fig. 1C, G, H; Fig. 4F).

3. The authors should include a discussion of how they think the peroxisome-PM tethering function of Inp1 relates to its peroxisome-ER tethering function; are they mutually exclusive, can Inp1 function in both tethering roles simultaneously? Such discussion would be greatly strengthened by the addition of experimental evidence that can be used to determine if one or both of the tethering functions of Inp1 is required for peroxisome retention.

We have now strengthened our conclusion that the plasma membrane binding domain of Inp1 is required for peroxisome retention and that this is the function of Inp1 in its role as a PM-PER tether. We also discuss two models by which Inp1 could function in both tethering roles simultaneously (a MECA-style model) or that peroxisome-ER tethering could occur independent of PM-PER tethering by Inp1.

Minor comments:

1. It would be helpful if amino acid numbers were added to the Inp1 schematic in Fig. 1C.

Amino acid numbers are now present on the Inp1 schematic in Fig. 2C (Formally Fig. 1C).

2. For the cell images shown, it should be stated if they are maximum intensity projects or single focal planes.

This has now been stated where relevant.

3. The legend for Fig. 3A and B states "mating pairs were scored..." but I believe it should be budding cells were scored.

For this assay, it was mating cell pairs that are forming a zygote which were scored. This has now been made clear in the figure legend.

Reviewer #2:

The authors would like to thank this reviewer for their constructive comments which we have implemented in our improved manuscript.

1. The liposome binding studies lack controls to show that the binding is specific and is not simply driven by charge. Other PIPs or lipids with equivalent charge should be used.

We have now carried out assays with liposomes containing PS to rule out that PI(4,5)P2 interaction is not due to a negative charge effect (Fig. 5F, G). As the reviewer will see, MBP-Inp1 does not interact with PC/PE liposomes supplemented with either PI(4)P or PS.

2. The competition of Inp1 and Pex19 for Pex3 binding is interesting, but it is not clear that it has any functional relevance. Does Pex19 binding to Pex3 regulate contact site formation?

We wanted to include the data regarding the competition between Inp1 and Pex19 as it shows a real molecular characterisation of the Inp1-Pex3 interaction. We have not explored in this study if Pex19 binding to Pex3 regulates contact site formation.

Reviewer #3

The authors would like to thank this reviewer for their extensive and constructive comments. We have implemented all of them in our revised and improved manuscript.

• In Figure 1A, it is not shown that the *Inp1* puncta necessary correspond to PO. This is, however, shown in Figure S1A - this is important so it might be clearer if this was moved to the main Figure?

That the puncta seen in Figure 2A (formerly Fig. 1A) are peroxisomes is clearly shown by the fact that they are absent in *pex3Δ* cells. In order to keep the figures as concise as possible, we have kept the data in Fig. S1A as it is and clearly referred to it in the text.

• Western blots for the various protein tags are not consistently shown for the *in vitro* binding experiments. Although the experiments are well-controlled, blots would be useful to confirm that the bands seen on the Coomassie gels are what they are identified as.

Protein levels were sufficient in the *in vitro* binding assays that Western blotting was not required. We are confident in our identification of the bands in the Coomassie gels as the MBP/GST-*Inp1* bands, moreover their breakdown products, resemble those published previously in Coomassie gels by Munck et al., 2009 and Western blots by Knoblach et al., 2013. Additionally the deletion of the LXXLL motif in GST-*Inp1* results in a full length band which migrates slightly lower than full length GST-*Inp1* in Fig. 2G, as would be expected. We also provide evidence that the band corresponding to His-Pex3 is identified correctly with the Western blots in Fig. 2B.

• In the competition binding assay, the authors show that only the C terminal of *Inp1* can successfully compete for Pex3 binding with Pex19. Since the authors identify the LXXLL motif as the binding site, does a recombinant version of their LL>AA *Inp1* mutant fail to compete with Pex19? Do the authors consider that this *Inp1*/Pex19 competition might be physiologically relevant?

In our *in vitro* competition assay we show that only the *Inp1* truncation which is able to bind to Pex3 (the C-terminus) can out compete Pex19 for binding. Additionally, we show in two separate *in vivo* experiments that the LL>AA *Inp1* mutant does not compete with Pex19 (Fig. 3C, D) As per the editor's comments, we feel that this experiment does not need to be further addressed.

We do think that the competition between the two proteins could be physiologically relevant, perhaps regulation of Pex3, Pex19 or Inp1 is required to ensure a balance between peroxisome retention and peroxisomal biogenesis. This is not something that was addressed in this study but could form the basis of future work.

• In the section on Inp1/Pex19 competition for Pex3 binding, the rationale for using the $\Delta mdh3/\Delta gpd1$ mutant is not clear - the experiment needs to be explained better. What does this reveal about Inp1 function?

The experiment using the *mdh3/gpd1* Δ mutant has been removed from the updated manuscript. It was included to emphasise the fact that overexpression of Inp1 leads to a loss of functional peroxisomes, however, the authors deemed this experiment to be surplus to requirement in the updated manuscript.

• The authors state that: "The absence of PO in cells overexpressing Inp1 is initially a result of over-retention in mother cells with buds failing to inherit POs". Do they have any evidence to support this e.g. a time-course during division, or is there a suitable reference?

A suitable reference has now been included for this phenotype, Fig. S1B of Munck et al., 2009.

• In Figure 3A, the authors say they observe PO retention in the mother cells. How are the mother and daughter cells distinguished? In addition, this panel shows a greater number of PTS1 puncta (= PO?) in cells in which POs are retained - why does this occur? This needs to be explained as it confounds interpretation.

Cells which were actively budding with small to medium sized buds were used for quantification and as the bud is smaller than the mother cell, we are able to determine between the two. In cells where there is a retention defect, it is clear which cell is the bud as all the peroxisomes are present there. The presence of fewer peroxisomal puncta in $\Delta inp1$ cells (which frequently form clusters as seen in Fig. 1A). compared to when peroxisomes are retained is well documented but not investigated.

• The authors provide a hypothesis for the different PO clustering seen when the minimal Inp1 tether is expressed, but this is somewhat confusing and needs to be explained better (there is a clearer explanation later in the manuscript).

After extensively re-writing the manuscript, we removed this experiment in order to streamline the data as we feel it no longer contributes to the main message of our study.

- ***The experiments shown in Fig 4 are very nice, but the interaction of MBP-Inp1 with PI(4,5)P2 is not quantified, unlike the others - this should be added.***

Quantification of MBP-Inp1 binding to PI(4,5)P2 has now been included.

- ***The experiments showing Inp1 and PO localisation at the PM are probably the weakest, as a result of the inherent confounding factor of cortical ER positioning. The authors have come up with the creative solution of using a mutant lacking cortical ER tubules, but it would be necessary to show if PO retention is normal in the mutant for this to be a suitable model system. This would perhaps also inform the ER-PO vs PM-PO issue. Also, only one example is shown (Fig. 5B) of Inp1 localising to the side of the PO next to the periphery and away from the ER - how typical was this localisation? The authors might also like to consider using a technique with sufficient resolution to identify PO-PM contacts in WT cells e.g. EM to demonstrate that these contacts form under physiological conditions.***

We have now shown that peroxisome retention is normal in both ER mutants and thus that they are suitable models. As disruption of the cortical ER does not appear to affect peroxisome retention, this points to the fact that additional peroxisomal cortical contact sites (e.g. with the plasma membrane) are involved. We have now included multiple examples throughout the paper where we show Inp1 localising to the periphery away from the ER and have included additional quantification of the observed phenotypes. The imaging we have carried out was of sufficient resolution to make our observations, using EM is not something we have considered as using EM to visualise peroxisome contact sites in glucose grown *S. cerevisiae* has been reported to be very difficult (Shai et al., 2018).

- ***In Figure 5B, the co-localised puncta of ER, PO and Inp1 are hard to see - enlarged ROIs would make this clearer. Does the inclusion of 3 z planes add anything - would the middle image not suffice?***

Enlarged ROIs have now been included in the revised manuscript with line scan analyses to further show the relative locations of proteins at the cell periphery. For Fig. 1G (formerly 5B), images of 2 consecutive z-planes are now included, this shows several peroxisomes which give representative examples of peroxisome localisation (e.g. at the end of an ER sheet, free of ER and sandwiched between the ER and cell periphery) more clearly than if a single focal plane is shown as not all the peroxisomes in the image are in the same focal plane. Additionally, the use of multiple z-planes illustrates that ER is really not in close proximity of the peroxisomes i.e. just underneath or above the peroxisome.

- ***Are the total number of POs the same in control and minimal-tether-overexpressing cells (Figure 5G)? If not, this could confound the observation that expression of the tether increases the number of PO-PM contacts.***

Overexpression of the minimal tether causes an over retention of peroxisomes in the mother cell as initial overexpression of wild type Inp1 does (see above point) but does not seem to affect overall peroxisome number or biogenesis, this is shown in Fig. S3D. More importantly, the minimal tether only affects the PM-PER reporter, and not any of the other reporters.

• Fig. 5G, PM-PER - should the PO not be associated with the PM more strongly (image 1) under these conditions?

The images shown are a compression of multiple z-stack images, meaning that peroxisomes appearing centrally in the cell may actually be at the top or bottom of the cell, associated with the PM. This was necessary to get an overview of the number of puncta per cell, using single planes would have underestimated the increase in contact site reporter puncta. This is now made clearer in the figure legend.

• The presentation of the data indicating that overexpression of the minimal tether increases the number of PM-PER puncta per cell (Figure 5H) is a little confusing and could be made clearer.

We acknowledge this point but feel that the graph shows all the data clearly by presenting the percentage of cells with X amount of reporter site puncta per cell in the control or with minimal tether overexpression. With tether overexpression there is a higher percentage of cells with 5+ reporter site puncta per cell than with the control.

• Would it be possible to design minimal tethers, or use mutant strains to separate Inp1's ER and PM binding functions, e.g. express Inp1 in which only PI(4,5)P₂ binding was disrupted? This would allow the relative contribution of each to a) contact site number and b) PO retention to be determined.

The first 100 amino acids of Inp1 localise to the plasma membrane (Fig. 5A), bind to PI(4,5)P₂ liposomes (Fig. 5), are sufficient for peroxisome relocation to the cell periphery in the absence of Inp1 (Fig. 4B) and are required for peroxisome retention as deletion of amino acids 1-100 results in a loss of retention in $\Delta inp1$ (Fig. 4A). We find no evidence that the N-terminus of Inp1 binds to Pex3 *in vitro* and as such we don't support the current model that Pex-ER tethering occurs via an Inp1/Pex3 bridge. We find no evidence that Inp1 1-100 interacts with the ER *in vivo* as overexpression of the minimal tether does not increase the PER-ER reporter puncta. Additionally, the minimal tether localises to areas of the cell periphery spatially resolved from the ER on the side of peroxisomal foci proximal to the plasma membrane. As such we believe this minimal tether comprises the plasma membrane binding domain of Inp1 which does not interact with the ER and that this functions in tethering peroxisomes to the cell cortex. As part of this study, we did find that when the first 52 amino acids of Inp1 are deleted in the endogenously expressed protein tagged with GFP (Inp1 52-420-GFP), peroxisome retention is rescued in $\Delta inp1$ cells. As Inp1 100-420-GFP does not rescue retention (Fig. 4A), this points to the fact that

retention relies on amino acids in the 53-100 region. However, when we attempted experiments with these smaller tethers, we found the proteins to be unstable and we were unable to make confident conclusions. We feel that it is perhaps too simplistic to simply shave off more and more amino acids in order to find an even more minimal tether. The plasma membrane and tethering ability of the N-terminal domain of Inp1 most likely relies on a tertiary folded structure and we can only speculate how the domain folds. Due to this, we felt it was most appropriate to use 1-100 as our minimal tether, as this is a protein which is stably expressed and behaves as such that we can make confident observations.

• The authors end by proposing the hypothesis that Inp1 tethers PO via dual interactions with the ER and the PM (although they have not addressed Inp1-dependent PO-ER contacts anywhere in their data). Do they imagine that peroxisomes could interact with both ER and PM at the same time? Can they speculate why such three-way positioning might be important? Contacts between one organelle with multiple others are an upcoming hot topic so some discussion here would be very interesting.

We have now addressed Inp1 dependent PO-ER contacts and in fact show that Inp1 is not required for peroxisomes to make contact with internal ER structures (Fig. 1D). We do not exclude that peroxisomes could interact with both the cortical ER and PM at the same time and have now included more discussion and proposed two models by which tethering to both membranes occurs via Inp1 or that additional cortical factors may be involved.

Throughout:

• No statistical analysis is presented anywhere and should be included. It is sometimes unclear how many independent experiments were performed.

We have stated where necessary how many independent experiments were performed and included SEM error bars in our quantitative experiments where appropriate.

March 29, 2020

Re: JCB manuscript #201906021R-A

Dr. Ewald H Hettema
University of Sheffield
Department of Molecular Biology and Biotechnology Firth Court
Western Bank
Sheffield S10 2TN
United Kingdom

Dear Dr. Hettema,

Thank you for submitting your revised manuscript entitled "The Pex3/Inp1 complex tethers yeast peroxisomes to the plasma membrane". The manuscript has been seen by the original reviewers whose full comments are appended below. While the reviewers continue to be overall positive about the work in terms of its suitability for JCB, some important issues remain.

We feel that it is particularly necessary to address the point about the charge of the liposomes brought up by both Rev#1 and #2, as the phospholipid specificity of Inp1 is a key part of the model proposed for the tethering of peroxisomes to the plasma membrane rather than the ER. Please also attend to the following formatting changes for resubmission:

- Provide the main and supplementary texts as separate, editable .doc or .docx files
- Provide main and supplementary figures as separate, editable files according to the instructions for authors on JCB's website paying particular attention to the guidelines for preparing images and blots at sufficient resolution for screening and production
- Provide tables as excel files
- Add scale bars to figures 1D (bottom row panel?), 1F (and zoomed in crop)

Our general policy is that papers are considered through only one revision cycle; however, given that the suggested changes are relatively minor we are open to one additional short round of revision. Please note that I will expect to make a final decision without additional reviewer input upon resubmission.

Please submit the final revision within one month, along with a cover letter that includes a point by point response to the remaining reviewer comments - but please let us know if you require longer because of the current pandemic.

Thank you for this interesting contribution to Journal of Cell Biology. You can contact me or the scientific editor listed below at the journal office with any questions, cellbio@rockefeller.edu.

Sincerely,

Jodi Nunnari, Ph.D.

Editor-in-Chief

Marie Anne O'Donnell, Ph.D.
Scientific Editor

Journal of Cell Biology

Reviewer #1 (Comments to the Authors (Required)):

The authors have addressed the majority of my concerns in the revised manuscript. They have included additional data that further support their conclusion that Inp1/Pex3 functions as a peroxisome-plasma membrane tether that is involved in the peroxisome retention in mother cells. The manuscript will be of interest to the cell biology community as it provides the first molecular description of a peroxisome-plasma membrane tether and successfully challenges the existing model that Inp1 functions in ER-peroxisome tethering and that ER-peroxisome tethering is required for peroxisome retention.

A few concerns need to be addressed prior to publication.

1. While the authors did add PS-containing liposomes to their in vitro lipid binding studies, the concentration of PS and PI4P used in the liposome sedimentation assays (22%) is equivalent to the concentration of PI4,5P2 used, which means that the PI4,5P2 liposomes still have a higher net negative charge. The assays need to be done in manner such that the net negative charge is equivalent. In other words, 44% PS or PI4P, 11% PI4,5P2, or a combination of 22% PS + 22% PI4P need to be used.
2. The authors need to be more careful with their descriptions that "the N-terminal 100 amino acids of Inp1 are necessary and sufficient for peroxisome retention" and "can act as a minimal tether that is both necessary and sufficient for peroxisome retention" in the last section of the results and discussion. They clearly demonstrate that the first 100 amino acids are necessary and sufficient for the interaction with the plasma membrane. However, while the first 100 amino acids are necessary for peroxisome retention, they are not sufficient for peroxisome retention. Amino acids 300-378 are required for Inp1's interaction with Pex3 and, therefore, the ability of Inp1 to interact with and retain peroxisomes.
3. The authors need to better clarify that the cell images shown in Fig. 5J are the same as those shown in Fig. S3F and that the lines they are showing in Fig. 5J correspond to the line scans shown in Fig. S3F.

Reviewer #2 (Comments to the Authors (Required)):

This study has been improved. It now makes a stronger case that the Inp1-Pex3 complex tethers peroxisomes to the PM and not the ER, as had been thought. As I said in my previous review, this finding, by itself, is only a modest advance. It was already known that the Inp1-Pex3 tether plays a role peroxisome inheritance, which this study confirms. There is no further conceptual advance. What, if anything, is the significance of the tethering beyond inheritance? How is the tethering regulated? To me, some headway on answering these questions seems necessary for this to be appropriate for JCB. I have some other, more minor concerns.

1. I am still not entirely convinced that Inp1 specifically binds PI(4,5)P2. The binding experiments use liposomes with 22% PI(4,5)P2, which is probably much higher than occurs in the PM. In addition, the control liposomes containing PI4P or PS are not charge balanced; PI(4,5)P2 has twice the charge of PI4P and PS.

2. Using *rtn1/rtn2/yop1* and delta-tether strains is good idea, but I have two concerns. First, instead of quantifying the percent of cells with peroxisomes at the cortex without ER, it would be better to know the percent of peroxisomes that are not in contact with ER and, of these, how many are next to the PM. Are these percentages different in when cells are also lacking Inp1? Second, it remains possible that some of the peroxisomes at the PM said to be free of ER have associated ER that can be seen in images above or below the focal plane shown (for example, in Fig. 1C.D). Since Z-stacks were taken, it would be good to show more of the stacks.

Reviewer #3 (Comments to the Authors (Required)):

The authors have included the vast majority of the reviewers' suggestions and have re-written the manuscript now providing more clarity and explanation. More insight into the relationship between ER and plasma membrane tethering has been provided. The new experimental data they present greatly improves the manuscript, as they have now provided convincing evidence that it is the Inp1-mediated peroxisome-plasma membrane tethering, and not the Inp1-mediated peroxisome-ER tethering, that regulates peroxisome retention. Furthermore, data on Inp1 and peroxisome localisation at the plasma membrane have been greatly improved. Overall, this is an important and interesting contribution, which provides novel mechanistic insight into peroxisome-plasma membrane tethering and its role in peroxisome inheritance in yeast.

Dear Professor Nunnari,

Thank you for considering our revised manuscript and allowing us an additional round of revision to address the minor amendments requested. We would also like to extend our thanks to the reviewers for their positive and thoughtful comments. Given the current global pandemic, we are grateful for the work of all involved in the fast processing of our manuscript.

Firstly, we have attended to the formatting changes as requested by the editor for our resubmission. Secondly, we have addressed all points raised by the reviewers. We will first give special attention to the points regarding the charge of the liposomes brought up by both Reviewer #1 and #2.

We hope that our explanation will take away your concerns and that you will consider our resubmission favourably.

Best wishes,

Ewald

Reviewer 1

1. While the authors did add PS-containing liposomes to their *in vitro* lipid binding studies, the concentration of PS and PI4P used in the liposome sedimentation assays (22%) is equivalent to the concentration of PI4,5P₂ used, which means that the PI4,5P₂ liposomes still have a higher net negative charge. The assays need to be done in manner such that the net negative charge is equivalent. In other words, 44% PS or PI4P, 11% PI4,5P₂, or a combination of 22% PS + 22% PI4P need to be used.

Reviewer 2

1. I am still not entirely convinced that Inp1 specifically binds PI(4,5)P₂. The binding experiments use liposomes with 22% PI(4,5)P₂, which is probably much higher than occurs in the PM. In addition, the control liposomes containing PI4P or PS are not charge balanced; PI(4,5)P₂ has twice the charge of PI4P and PS.

Firstly, it is important to state that altering the amount of PI(4,5)P₂ in the PC/PE liposomes, as suggested by Reviewer 1, would be a straightforward experiment for us to do. Unfortunately, as many other labs around the world, we have no access to our lab at this point or for the foreseeable future due to the ongoing pandemic and we are therefore unable to perform additional experiments at this time. We do however have additional experimental data which addresses the lipid charge concerns of both reviewers, albeit in a slightly different way. We have now included this data in our manuscript (Fig. 5F and S3G) and would like to take this opportunity to explain the rationale of our approach.

Our key *in vitro* observation is that Inp1 is a lipid binding protein, as determined by the Folch fraction 1 extract liposome binding study (Fig. 5B). We then tried to determine which lipids in this fraction Inp1 is binding to. We focused on the plasma

membrane lipids PS, PI(4,5)P₂ and the ER/secretory pathway lipid PI(4)P. Only incorporation of PI(4,5)P₂ into PC/PE liposomes resulted in clear binding of Inp1. We were surprised about the reviewer's concerns regarding charge balancing of the assays. We used molar balanced amounts of phospholipids as is generally used in these types of experiments (See, for instance Chandra et al., 2019, Kume et al., 2016; Ping et al., 2016). The Ayscough lab has published several studies since 2015 that use the lipid binding assays as described in our manuscript, and never had a request from reviewers to balance the charge.

We believe that our experiment to be a relevant comparison as:

1) Proteins can interact with phospholipid head groups by electrostatic interaction with the charged headgroup or by specific binding with the headgroup (Zhao and Lappalainen 2012; Lemmon 2008).

Both factors result in a degree of binding specificity and both could be physiologically relevant. Even if the specificity was entirely dependent on differences in charge of the headgroup, it would still provide specificity between PI(4,5)P₂ on one hand (i.e. plasma membrane binding) and PI(4)P (secretory pathway including ER) and PS on the other.

2) Our experiments are carried out in the presence of 20mM KCl + 160 mM NaCl, which would tend to minimise non-specific electrostatic interaction between lipid headgroups and binding protein (Zhao and Lappalainen 2012). We have now included experiments where we increased salt concentration further to 20mM KCl + 350mM NaCl (Fig. S3F and 5G). Loss of interaction at a higher salt concentration indicates that the interaction between a protein and lipid is mainly mediated through non-specific electrostatic interactions. MBP-Inp1 showed clear binding to PC/PE liposomes containing PI(4,5)P₂ in the presence of high salt (370mM) which shows that this interaction can take place even under high salt concentrations, albeit less efficiently than at 180mM salt. These results suggest that electrostatic interactions do contribute in binding of Inp1 to the liposomes but that there is specificity in Inp1 binding to PI(4,5)P₂ containing liposomes and the interaction is not based on charge only. This data gives us further confidence that the interaction we repeatedly see between MBP-Inp1 and PI(4,5)P₂ liposomes is not solely due to charge.

3) The estimated percentages of PI(4,5)P₂ and PI4P in membranes is thought to be approximately 0.05% in each case, so we consider that keeping the proportions equal in our experiments is appropriate.

4) Liposome binding assays are artificial representations of protein-membrane interactions. But, even in the presence of 22% PI(4)P, no binding of Inp1 is found above background binding. We do find a low (salt sensitive) amount of Inp1 cofractionating with PS containing liposomes, so increasing the level of PS may increase Inp1 binding. As PS is mainly concentrated and highly abundant in the plasma membrane, it does not affect our final conclusion that Inp1 binds to plasma membrane lipids. As we stated above, we are currently unable to perform these experiments as our labs are in lock down as a consequence of COVID-19.

5) Finally, our lipid binding studies are not stand alone experiments but in fact support a number of other experiments presented in this paper. Our *in vivo* experiments show that the N-terminal domain of Inp1 fused to GFP localises to the plasma membrane (Fig. 5A) and where plasma membrane levels of PI(4,5)P₂ levels are decreased in *sac1Δ* cells, this domain shows a decrease in PM localisation (Fig. 5J, S3G). Under these conditions, PI(4)P levels are increased (10-20 fold) and the PI(4)P accumulates at the ER and vacuole (Tahirovic et al., 2004). Nonetheless, we do not see Inp1 1-100-GFP recruitment to ER or vacuolar membranes in the *sac1Δ* mutant. Furthermore, when the Num1 PH domain, known to specifically bind PI(4,5)P₂, is targeted to peroxisomal membrane, it can substitute for Inp1 in peroxisome retention. We have several independent experiments that support a role for PI(4,5)P₂ in Inp1 association with the PM. We therefore conclude that our *in vivo* and *in vitro* data are in support of each other in showing that Inp1 binds preferentially to the plasma membrane lipid PI(4,5)P₂ and perhaps to some extent PS. None of our experiments indicate that Inp1 is able to bind PI(4)P.

We will now address the additional comments of the reviewers:

Reviewer #1 (Comments to the Authors (Required)):

1. While the authors did add PS-containing liposomes to their *in vitro* lipid binding studies, the concentration of PS and PI4P used in the liposome sedimentation assays (22%) is equivalent to the concentration of PI4,5P₂ used, which means that the PI4,5P₂ liposomes still have a higher net negative charge. The assays need to be done in manner such that the net negative charge is equivalent. In other words, 44% PS or PI4P, 11% PI4,5P₂, or a combination of 22% PS + 22% PI4P need to be used.

This has been addressed above.

2. The authors need to be more careful with their descriptions that "the N-terminal 100 amino acids of Inp1 are necessary and sufficient for peroxisome retention" and "can act as a minimal tether that is both necessary and sufficient for peroxisome retention" in the last section of the results and discussion. They clearly demonstrate that the first 100 amino acids are necessary and sufficient for the interaction with the plasma membrane. However, while the first 100 amino acids are necessary for peroxisome retention, they are not sufficient for peroxisome retention. Amino acids 300-378 are required for Inp1's interaction with Pex3 and, therefore, the ability of Inp1 to interact with and retain peroxisomes.

We agree with the reviewer and we have now changed the text and state that "the N-terminal 100 amino acids of Inp1 are necessary and sufficient for peroxisome retention" where this part of Inp1 is associated with peroxisomes.

3. The authors need to better clarify that the cell images shown in Fig. 5J are the same as those shown in Fig. S3F and that the lines they are showing in Fig. 5J correspond to the line scans shown in Fig. S3F.

This has now been addressed by amending the relevant figure legends.

Reviewer #2 (Comments to the Authors (Required)):

1. I am still not entirely convinced that Inp1 specifically binds PI(4,5)P₂. The binding experiments use liposomes with 22% PI(4,5)P₂, which is probably much higher than occurs in the PM. In addition, the control liposomes containing PI4P or PS are not charge balanced; PI(4,5)P₂ has twice the charge of PI4P and PS.

This has been addressed above. Since, we have not tested any other lipids than PC, PE, PI(4)P, PI(4,5)P₂ and PS we can't rule out that Inp1 binds other lipids and is solely specific for PI(4,5)P₂. We have amended the text throughout the manuscript and changed 'the N-terminal PI(4,5)P₂ binding domain' into an N-terminal lipid binding domain that binds PI(4,5)P₂.

2. Using rtn1/rtn2/yop1 and delta-tether strains is good idea, but I have two concerns. First, instead of quantifying the percent of cells with peroxisomes at the cortex without ER, it would be better to know the percent of peroxisomes that are not in contact with ER and, of these, how many are next to the PM. Are these percentages different in when cells are also lacking Inp1? Second, it remains possible that some of the peroxisomes at the PM said to be free of ER have associated ER that can be seen in images above or below the focal plane shown (for example, in Fig. 1C.D). Since Z-stacks were taken, it would be good to show more of the stacks.

Whilst we acknowledge the point of the reviewer, we felt that quantifying the percent of cells with peroxisomes at the cortex without ER was a better way of presenting our data as it incorporates a large number of cells (n=150 and 170). This gives an accurate representation of what is seen in the cell population as a whole rather than individual cells. Additional quantification of this data is not possible at this time due to us not having access to our lab and therefore not having access to our microscope data. Furthermore, changing the way we quantified this phenotype would not alter our conclusions in any way, we would still find peroxisomes at the cortex without the ER which is the observation we have made here. With regards to the second point, for a peroxisome to be considered free of ER it was ensured during the quantification process that the ER was not present above or below the focal plane. We did show an example of images above and below the focal plane in our original submission but reduced this to two consecutive z-stack planes on the advice on Reviewer 3. Figure 1G shows two consecutive z-stack planes.

Reviewer #3 (Comments to the Authors (Required)):

No points to address.

References from this letter:

References from this letter:

- 1) Lemmon MA. Membrane recognition by phospholipid-binding domains. *Nature Reviews Mol. Cell Biol.* 9;99-111 (2008).
- 2) Zhao X. and Lappalainen P. A simple guide to biochemical approaches for analyzing protein–lipid interactions. *Mol. Biol. Cell*, 23;2823-30 (2012).
- 3) Chandra, M., Chin, Y.K., Mas, C. *et al.* Classification of the human phox homology (PX) domains based on their phosphoinositide binding specificities. *Nat Commun* 10, 1528 (2019).
- 4) Kume, A et al., The Pleckstrin Homology Domain of Diacylglycerol Kinase η Strongly and Selectively Binds to Phosphatidylinositol 4,5-Bisphosphate. *Journal of Biological Chem.* 291;8150-61 (2016).
- 5) Ping HA, Kraft LM, Chen W, Nilles AE, Lackner LL. Num1 anchors mitochondria to the plasma membrane via two domains with different lipid binding specificities. *J. Cell Biol.* 213:513-24 (2016).
- 6) Tahirovic, S., Schorr, M., Mayinger, P. Regulation of Intracellular Phosphatidylinositol-4-Phosphate by the Sac1 Lipid Phosphatase. *Traffic* 6:116-30 (2004).

Protein-Liposome binding studies by the Ayscough lab

- 1) Smaczynska-de Rooij II, Marklew CJ, Palmer SE, Allwood EG, Ayscough KR. Mutation of key lysine residues in the Insert B region of the yeast dynamin Vps1 disrupts lipid binding and causes defects in endocytosis. *PLoS One.* 14:e0215102 (2019).
- 2) Rzepnikowska W, Flis K, Kaminska J, Grynberg M, Urbanek A, Ayscough KR, Zoladek T. Amino acid substitution equivalent to human chorea-acanthocytosis I2771R in yeast Vps13 protein affects its binding to phosphatidylinositol 3-phosphate. *Hum Mol Genet.* 26:1497-1510 (2017).
- 3) Urbanek AN, Allwood EG, Smith AP, Booth WI, Ayscough KR. Distinct Actin and Lipid Binding Sites in Ysc84 Are Required during Early Stages of Yeast Endocytosis. *PLoS One.*10:e0136732 (2015).